# WNT signalling promotes NF-κB activation and drug resistance in KRAS-mutant colorectal cancer

Bojie Cong[1,2], Evangelia Stamou[1], Kathryn Pennel [ID][1], Teena Thakur[1,3], Molly Mckenzie[1], Amna Matly[1], Kathryn Gilroy [ID][3], Harshit Shah[1,3], Sindhura Gopinath[4], Joanne Edwards [ID][1] & Ross Cagan [ID][1✉]

## Abstract

**Approximately 40% of colorectal cancer (CRC) cases are characterised by KRAS mutations, rendering them insensitive to most therapies. While the reasons for this resistance remain incompletely understood, one key aspect is genetic complexity: in CRC, oncogenic KRAS is most commonly paired with mutations that alter WNT and P53 activities ("RAP"). Here, we demonstrate that elevated WNT activity upregulates canonical NF-κB signalling in both _Drosophila_ and human RAS mutant tumours. This upregulation was enhanced by P53 loss and required immune-associated factors Toll-1 and Toll-9. These changes reduced efficacy of Ras pathway-targeting drugs such as trametinib due to NF-κB-dependent enhancement of the glucuronidation detoxifying pathway, likely through modulating gene transcription and glucose uptake. Inhibiting WNT activity pharmacologically suppressed trametinib resistance in RAP tumours and more genetically complex 'patient avatar' models. The efficacy of WNT/MEK drug inhibitor combinations was further enhanced by targeting _brm_, _shg_, _ago_, _rho-GAPp190_, and _upf1_, potential biomarkers for patients responsive to this dual therapeutic approach. These findings shed light on how genetic complexity impacts drug resistance and a strategy to overcome it.**

**Keywords** Colorectal Cancer; _Drosophila_; Glucuronidation; NF-κB; WNT
**Subject Categories** Cancer; Metabolism; Neuroscience

## Introduction

Colorectal cancer (CRC) remains a leading cause of cancer-related deaths worldwide. Progressive disease is characterised by uncontrolled cell growth, primarily driven by a complex array of genetic mutations (Sung et al, 2021; Stratton et al, 2009; Stratton, 2011). CRC cases featuring mutations in the RAS family of small GTPases have proven especially problematic due to a striking insensitivity to most targeted therapies in the clinics (Yaeger et al, 2018; Cremolini et al, 2015; Wang et al, 2022). Despite the development of successive generations of inhibitors targeting the RAS pathway that demonstrate promise in pre-clinical studies, these inhibitors have shown minimal or transient activity in RAS-mutant CRC patients. While resistance mechanisms such as elevated KRAS or EGFR activity can lead to emergent drug resistance in some tumours (Huang et al, 2021; Hofmann et al, 2022; Zhang et al, 2018), the primary mode(s) of resistance for many CRC tumours remain unclear.

One source of drug resistance is genetic complexity, a consistent and key hallmark of CRC (Caponigro and Sellers, 2011; Bangi et al, 2016; Wang et al, 2022). Experiments in Drosophila and mouse models report that oncogenic RAS/KRAS mutations alone are only sufficient to initiate benign tumours (Pagliarini and Xu, 2003; Calcagno et al, 2008). The progression to malignancy in human CRC requires acquisition of additional mutations, most commonly in _P53_ and the WNT regulator _APC_ (Fearon and Vogelstein, 1990; Boutin et al, 2017; Bangi et al, 2016). Our recent studies involving _Drosophila_ CRC models have also demonstrated that genetic complexity amplifies metastatic potential and, key to this report, fosters drug resistance (Bangi et al, 2016). The precise mechanisms that connect genetic complexity to drug resistance remain poorly understood.

Here, we examine the impact of mutated _apc_ and _p53_ on Ras[G12V]-expressing tumours in the _Drosophila_ hindgut. We found that elevated WNT (Wg in _Drosophila_) activity led to upregulation of canonical nuclear factor-kappa B (NF-κB) signalling in Ras[G12V] tumours, which in turn led to emergent resistance to drugs such as the MEK inhibitor trametinib by elevating glucuronidation pathway activity. This resistance was reversed by reducing WNT activity with inhibitor compounds PNU-74654 or LF3, restoring trametinib-mediated rescue. P53 also contributed to drug response by regulating tumour growth and NF-κB signalling in high Wnt /Ras[G12V] hindgut tumours. Consistent with this data, human CRC samples with high WNT activity and oncogenic KRAS were associated with elevated canonical NF-κB signalling. Together, our data identify NF-κB as a key mediator of drug resistance in RAS-WNT-P53 CRC tumours, and suggest a novel approach to the treatment of KRAS-mutant CRC.

[1]School of Cancer Sciences, University of Glasgow, Wolfson Wohl Cancer Research Centre; Garscube Estate, Switchback Road, Bearsden, Glasgow, Scotland G61 1QH, UK. [2]Department of Biopharmaceutical Sciences, College of Pharmacy, Harbin Medical University, Harbin 150081, China. [3]CRUK Beatson Institute, Garscube Estate, Switchback Road, Glasgow, Scotland G61 1BD, UK. [4]Department of Cell, Developmental and Regenerative Biology, Icahn School of Medicine at Mount Sinai, 25-82 Annenberg Building; Box 1020, One Gustave L. Levy Place, New York, NY 10029, USA. ✉E-mail: Ross.Cagan@glasgow.ac.uk

# Results

## Wnt signalling reduced trametinib efficacy by elevating canonical NF-κB signalling

RAS family proteins regulate key cellular processes through multiple pathways, including the canonical Raf-MEK-ERK (MAPK) signalling pathway. The FDA-approved drug trametinib is a potent and precise MEK inhibitor that, despite strong preclinical activity, failed to demonstrate significant efficacy in CRC patients (Nalli et al, 2021; Infante et al, 2012). Consistent with these reports, we observed that trametinib—delivered orally in the food—strongly rescued tumour-induced lethality in *Drosophila* CRC models that target Ras$^{G12V}$ to the hindgut (*Ras$^{G12V}$*; Fig. 1A). However, trametinib failed to rescue a multi-targeting CRC model that combined oncogenic *Ras$^{G12V}$* plus RNAi ("i") knockdown of *Apc$^i$* and *P53$^i$* (*RAP*; Fig. 1A; all transgenes in this study are targeted to the hindgut via the *byn-GAL4* transcriptional driver). Trametinib did not affect control animals (Fig. EV1A). These data indicate an emergent resistance to trametinib in *RAP* hindgut tumours.

In a screen for pathways such as ABC transporters, JNK signalling and autophagy that distinguish *Ras$^{G12V}$* and *RAP* models, we found that canonical NF-κB pathway activity—also known as the Toll pathway in *Drosophila* (Meng et al, 1999) was increased in *RAP* models (Fig. 1B). Previous studies have shown that inhibition of NF-κB signalling increases sensitivity of HTC15 human colon cancer cells to the chemotherapeutic daunomycin by modulating drug uptake (Bentires-Alj et al, 2003). Interestingly, we found that inhibition of canonical NF-κB signalling by targeted knockdown of NF-κB family members dorsal-related immunity factor (*dif*) or dorsal (*dl*) (Minakhina and Steward, 2006) strongly improved trametinib's ability to rescue *RAP*-induced lethality (Fig. 1C). Knockdown did not affect the survival of control animals or *RAP* animals in the absence of trametinib (Fig. 1C,D). Conversely, elevating canonical NF-κB activity by targeted knockdown of the NF-κB inhibitor *cactus* (*cact*, an IκB orthologue) (Geisler et al, 1992; Kidd, 1992) reduced trametinib efficacy in *Ras$^{G12V}$* tumours (Fig. 1E). Knockdown of *cact* did not affect lethality in the absence of trametinib or in control animals (Fig. 1E,F), indicating the impact of NF-κB was trametinib-specific.

In our previous study, we reported that cooperative activation of Wnt and Ras led to trametinib resistance by enhancing glucose uptake in a Pi3k/Akt dependent manner (Cong et al, 2025). Here, we observed that the canonical NF-κB pathway reporter *drosomycin* was elevated when Ras and Wnt activities were elevated together in the hindgut but not when either gene was activated alone (Fig. 1G). These data indicate that Wnt activity induces trametinib resistance at least in part by elevating canonical NF-κB signalling in *Ras$^{G12V}$* tumours. We also observed that loss of function *p53* increased the level of canonical NF-κB signalling (compare Fig. 1B to Fig. 1G), suggesting that *p53* also contributes to the regulation of canonical NF-κB signalling in *RAP* tumours.

To gain a deeper understanding of the drug impact on *Ras$^{G12V}$* vs. *RAP* tumours, we assessed the effect of trametinib on tumour growth in the *Drosophila* hindgut proliferative zone (HPZ). Trametinib exhibited a near-complete suppression of HPZ tumour overgrowth in *Ras$^{G12V}$* tumours (Fig. EV1B,E–G). However, trametinib only partially suppressed HPZ tumour overgrowth of *RAP* tumours (compare Fig. EV1K to Fig. 1H, quantified in 1K). Inhibition of canonical NF-κB activity by knockdown of *dl* or *dif*

did not significantly suppress *RAP* tumour overgrowth in the presence of trametinib (compare Fig. 1H to 1I,J, quantified in 1K). Moreover, elevating Wnt signalling did not alter tumour overgrowth in *Ras$^{G12V}$* tumours or control animals (compare Fig. EV1H,I to EV1F,E, quantified in EV1C). However, in *byn-Ras$^{G12V}$* tumours, HPZ overgrowth was significantly enhanced by both elevated Wnt activity and, more weakly, by loss of p53 (compare Fig. EV1J to EV1F,H, quantified in EV1D). Together, these data indicate that (i) Wnt activation and p53 loss cooperate in promoting overgrowth in *Ras$^{G12V}$* tumours and (ii) canonical NF-κB signalling reduces trametinib efficacy by primarily modulating host survival rather than tumour growth in *Drosophila*.

Of note, we observed that elevated Wnt signalling led to melanisation of *Ras$^{G12V}$* tumours (compare Fig. EV1I to EV1F or EV1H); melanisation also was observed in *RAP* tumours (Fig. EV1K). Melanisation is an immune response triggered locally by injury, suggesting that co-activation of Wnt and Ras contributes to a 'wound-like' response in the hindgut. Previous work reported that local wounding led to increased canonical NF-κB activity in *Drosophila* (Capilla et al, 2017), consistent with a model in which elevated Wnt signalling leads to a wound response, upregulation of canonical NF-κB activity, and emergent drug resistance in *Ras$^{G12V}$* tumours.

## Toll-1 and Toll-9 are required for upregulation of NF-κB activity in RAP tumours

Toll-like receptors activate canonical NF-κB signalling by binding their ligands (Imler and Hoffmann, 2001). In *RAP* flies, knockdown of Toll-1 and Toll-9 robustly enhanced trametinib rescue of tumour-induced lethality 48 and 43%, respectively, in the presence of trametinib; knockdown of Toll-6 more weakly (29%) impacted trametinib response, while Toll proteins 3, 4, 5, 7 and 8 had no significant effect (Figs. 2A and EV2A). Importantly, knockdown of Toll-1 or Toll-9 did not significantly affect tumour-induced lethality in the absence of trametinib or control animals in the presence of trametinib (Fig. 2A,B), mirroring our results with NF-κB.

These data prompted us to test whether Toll-1 and Toll-9 are essential for inducing canonical NF-κB activity in *RAP* tumours. A recent study has demonstrated that Toll-9 can elevate canonical NF-κB activity to control proliferation and apoptosis in *Drosophila* imaginal discs in a Toll-1-dependent manner (Shields et al, 2022). We observed that nuclear translocation of Dorsal was strongly increased in *RAP* tumours compared to control (Fig. 2C,D), an indication of elevated pathway activity. Canonical NF-κB activity was suppressed by knockdown of Toll-1 or Toll-9 in *RAP* tumours (compare Fig. 2E,F to Fig. 2D), indicating that Toll-1 and Toll-9 is indeed required for enhancing canonical NF-κB activity in the context of *RAP* hindgut tumours. Also consistent with the results above, knockdown of Toll-1 or Toll-9 did not significantly suppress *RAP* hindgut proliferation in the presence of trametinib (compare Figs. 1H to EV2B,C, quantified in EV2D).

## Canonical NF-κB activity impacted trametinib response by altering its glucuronidation

Our data suggest a model in which combined alterations in Ras, APC, and P53 lead to Toll-mediated activation of canonical NF-κB

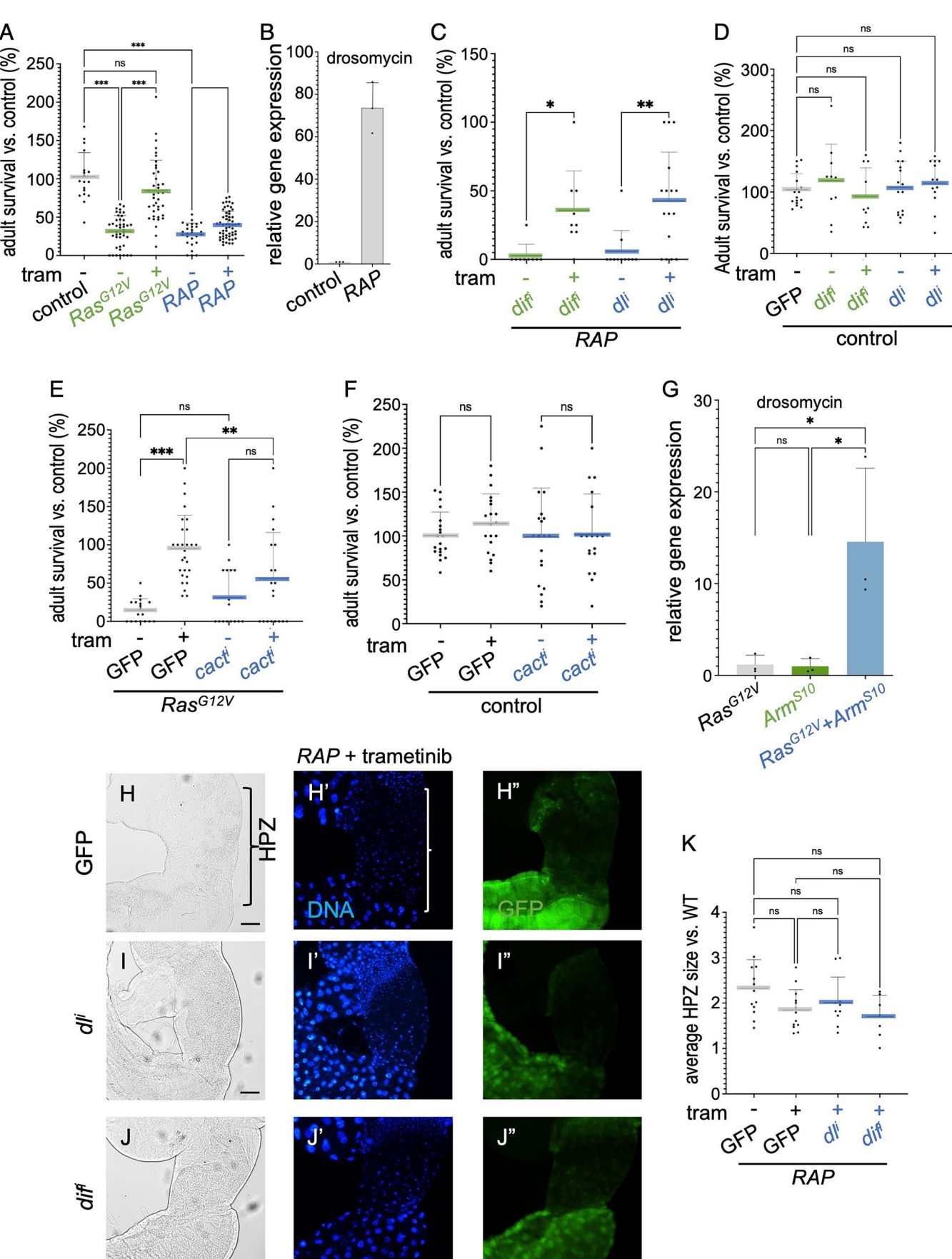

Figure 1. WNT signalling induced trametinib resistance through enhancing canonical NF-κB signalling.

(A) Percent survival of transgenic flies to adulthood relative to control flies was quantified in the presence or absence of trametinib (1 μM). Control (DMSO, n = 18), Ras^G12V (DMSO, n = 42; tram, n = 39), RAP (DMSO, n = 27; tram, n = 56). (B) Expression levels of *drosomycin* were quantified for each genotype by quantitative RT-PCR. Control (n = 3) and RAP (n = 3). (C–F) Percent survival of transgenic flies to adulthood relative to control flies was quantified in the presence or absence of trametinib (1 μM). (C) RAP;dif^i (DMSO, n = 9; tram, n = 9) and RAP;dl^i (DMSO, n = 11; tram, n = 16); (D) Control (DMSO, n = 17), dif^i (DMSO, n = 10; tram, n = 10) and dl^i (DMSO, n = 16; tram, n = 13); (E) Ras^G12V;GFP (control (DMSO, n = 18; tram, n = 28)) and Ras^G12V;cact^i (DMSO, n = 16; tram, n = 20); (F) GFP (DMSO, n = 20; tram, n = 20) and cact^i (DMSO, n = 20; tram, n = 18). (G) Expression levels of *drosomycin* were quantified for each genotype by quantitative RT-PCR. Ras^G12V (n = 3), arm^S10 (n = 3), and Ras^G12V;Arm^S10 (n = 3). (H–J) Images of the digestive tract of third instar larvae in the presence of trametinib (1 μM), which include the hindgut proliferation zone (HPZ). Nuclei are visualised with 4′,6-diamidino-2-phenylindole (DAPI) staining, and the hindgut is visualised by GFP. Scale bar 200 μm. (K) The average size of the hindgut proliferation zone (HPZ) size was measured by Fiji ImageJ and quantified as a relative size to the wild-type hindgut. All transgenes were expressed in the hindgut using byn-GAL4. RAP (DMSO, n = 13; tram, n = 13), RAP;dl^i (tram, n = 10), RAP;dif^i (tram, n = 7). (A, B, E–J) The experiments were conducted at 27 °C. (C, D) The experiments were conducted at 29 °C. The statistical tests used to calculate the P value are as follows: (A, C–G, K) one-way ANOVA; NS P(>0.12), *P(0.033), **P(0.002) and ***P(<0.001). All statistical data are summarised in Table EV1. The error bar is a standard deviation (SD), with each point representing biological replicates and numbers (n), including three technical replicates. Source data are available online for this figure.

signalling, which in turn promotes resistance to trametinib. To better understand the link between NF-κB activity and drug resistance, we used RNA sequencing (RNA-Seq) analysis to compare RAP;dl^i and RAP tumours treated with trametinib. We identified 1032 significantly altered genes: 446 downregulated and 566 upregulated in RAP;dl^i tumours when compared to RAP tumours (Fig. 3A and Dataset EV1; padj.<0.05, |log2(fold change)| >0.3).

Knockdown of dl led to (i) a reduction in Toll signalling targets, including drs, atta and wntd as expected. Reduced dl also led to reduced Imd-associated immune signalling (dptb, pirk), as well as signalling pathways Jak/Stat (upd1, upd2, upd3, dome socs36e) and Jnk (kay, puc, mmp1; Fig. 3B and Dataset EV1). However, reducing activity of Imd (relish^i), Jak/Stat (domeless^i) or Jnk (basket^i) did not significantly reduce trametinib resistance in RAP tumours (Fig. EV3A). We therefore focused on other factors highlighted by our RNA-Seq analysis.

Recently, we reported that RAP tumours displayed emergent upregulation of the glucuronidation pathway, a detoxification pathway known to directly inactivate many cancer drugs, including trametinib (Cong et al, 2025). We further demonstrated that the glucuronidation pathway was enhanced by the pentose phosphate pathway, linking detoxification to circulating glucose (Cong et al, 2025). Here, we observed that reduced dl (RAP;dl^i) led to reduced expression of key glucuronidation pathway enzymes, including sgl (human ortholog: UGDH) as well as pentose phosphate pathway enzymes, including pgd (PGD), rpi (RPIA), and zw (G6PD; Fig. 3B; Dataset EV1). Further, knockdown of dl significantly suppressed levels of glucuronidated trametinib in RAP;dl^i tumours as assessed by UDP release (Fig. 3C). This data indicates that canonical NF-κB signalling promotes resistance to trametinib at least in part by regulating genes that control glucuronidation, in turn helping tumours directly inactivate and expel drugs.

To identify novel canonical NF-κB downstream factors that influence trametinib response, we built on results from our RNA-Seq screen by performing a focused RNAi-based screen of 45 genes, comparing RAP vs. Ras^G12V tumours. RAP response to trametinib was strongly impacted by knockdown of blanks (human orthologue ADARB1), chitinase 4 (cht4, human orthologue AMCase or CHIA), major facilitator superfamily transporter 14 (mfs14, human orthologues SLC17A2, A3, A5, A7 and A8); chitinase 5 (cht5, human orthologue CHIA) weakly impacted trametinib response in RAP tumours (Figs. 3D,E; EV3B–E). Of note, knockdown of blanks,

cht4 or mfs14 did not affect control flies in the presence of trametinib (Fig. 3F). Knockdown of cht4 did not affect tumour development in the absence of trametinib, while blanks and mfs14 weakly affected tumour development (compare Fig. 3D to EV3F): RAP alone displayed 7% survival compared to RAP;cht4^i (1%), RAP;blanks^i (17%) and RAP;mfs14^i (23%). Also, in the context of RAP, knockdown of dl strongly reduced blanks, cht4 expression and, more weakly, mfs14 and cht5 (Fig. 3G).

Previous work has shown that overexpression of the human cht4 orthologue CHIA upregulated PI3K/AKT activation in human lung epithelial cells in vitro (Hartl et al, 2009). Our previous study in Drosophila showed that elevated Pi3k/Akt signalling enhanced glucuronidation by increasing glucose uptake in RAP tumours, and the level of trametinib glucuronidation can be measured by the released UDP levels (Cong et al, 2025). Here, we found that knockdown of cht4 significantly suppressed the level of released UDP in the presence of trametinib in RAP tumours (Fig. 3H), suggesting that canonical NF-κB activity also influences trametinib response by regulating Pi3k-mediated glucose uptake.

To assess canonical NF-κB pathway signalling as a therapeutic target, we removed one functional copy of dl in the context of the whole animal using a dl[1] loss of function allele (RAP;dl^{1/+}). Subtle reduction of dl was sufficient to strongly enhance the ability of trametinib to reduce RAP lethality (Fig. EV3F), supporting canonical NF-κB pathway components as adjunct therapies. Administering NF-κB inhibitors QNZ (EVP4593) or JSH-23 reduced NF-κB pathway activity as assessed with drosomycin expression (Fig. EV3G–J). JSH-23 trend towards improved survival did not rise to significance, while QNZ reduced survival: in addition to reducing NF-κB pathway activity, QNZ also inhibits production of TNF-α (Tobe et al, 2003), suggesting that off-targets may contribute to reduced survival.

## Wnt inhibitors increased trametinib efficacy on RAP tumours

Our results indicate that Wnt activation plays a role in regulating host lethality by increasing canonical NF-κB activity and, in turn, reducing trametinib's ability to suppress tumour progression in Ras^G12V animals. We therefore next assessed whether inhibiting WNT activity would reverse drug resistance in RAP tumours by feeding RAP flies one of several (Figs. 4A and EV4A–D) WNT inhibitors plus trametinib. The result was emergent rescue: in

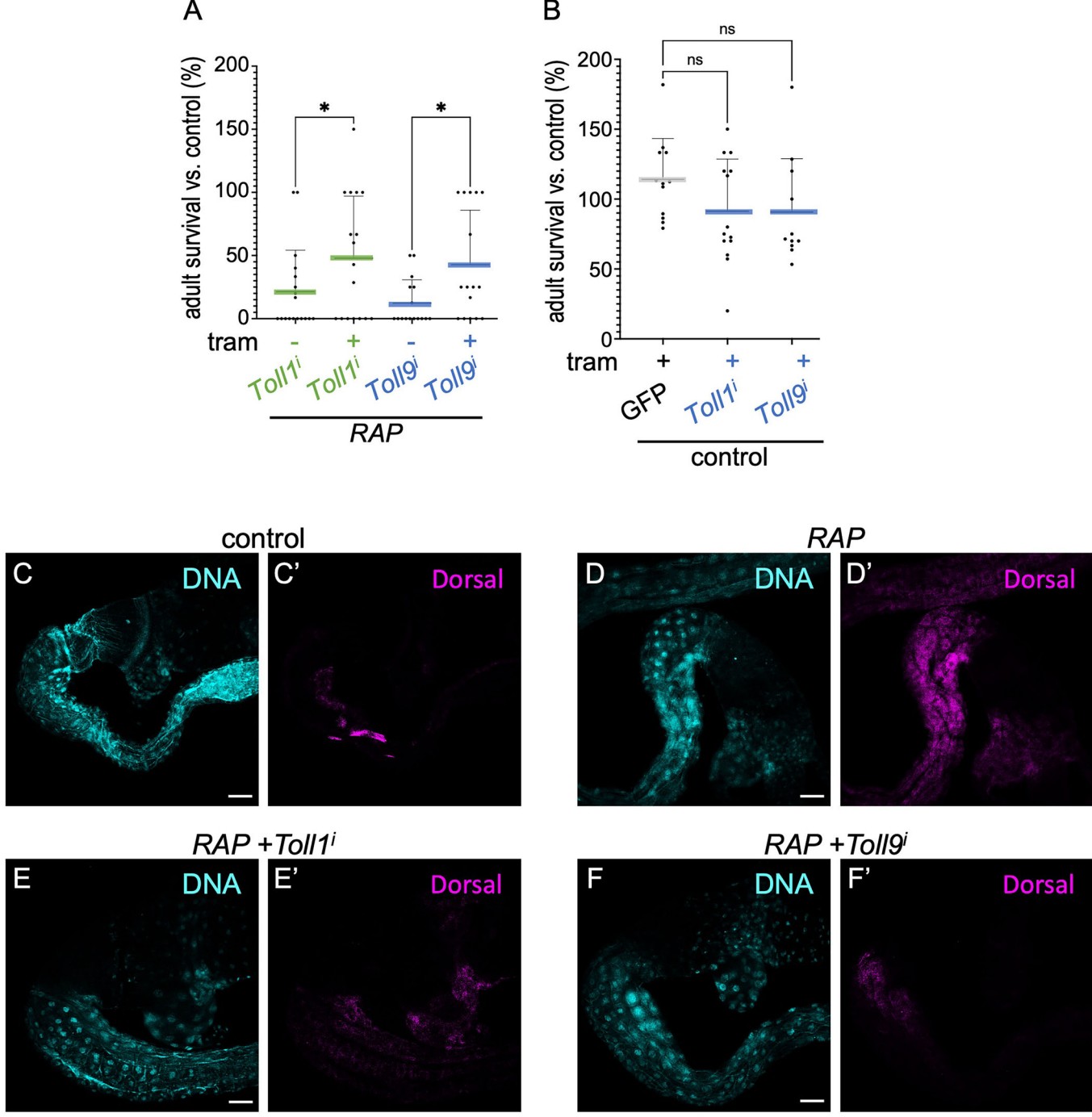

**Figure 2. Toll-1 and Toll-9 are required for upregulation of canonical NF-κB activity in *RAP* tumours.**

(A, B) Percent survival of transgenic flies to adulthood relative to control flies was quantified in the presence or absence of trametinib (1 μM). (A) *RAP*;*Toll1^i^* (DMSO, $n = 18$; tram, $n = 17$) and *RAP*;*Toll9^i^* (DMSO, $n = 16$; tram, $n = 16$); (B) Control (tram, $n = 12$), *Toll1^i^* (tram, $n = 14$) and *Toll9^i^* (tram, $n = 11$). (C–F) Control (C), *RAP* (D), *RAP*;*Toll1^i^* (E), *RAP*;*Toll9^i^* (F) transgenes were induced in hindguts and were stained with anti-dorsal antibody. DNA were visualised with propidium iodide (PI), Scale bar 200 μm. (A–F) The experiment was conducted at 29 °C. The statistical tests used to calculate the *P* value are as follows: (A, B) one-way ANOVA; NS $P(>0.12)$, *$P(0.033)$, **$P(0.002)$ and ***$P(<0.001)$. All statistical data are summarised in Table EV1. The error bar is a standard deviation (SD), with each point representing biological replicates and numbers (*n*), including three technical replicates. Source data are available online for this figure.

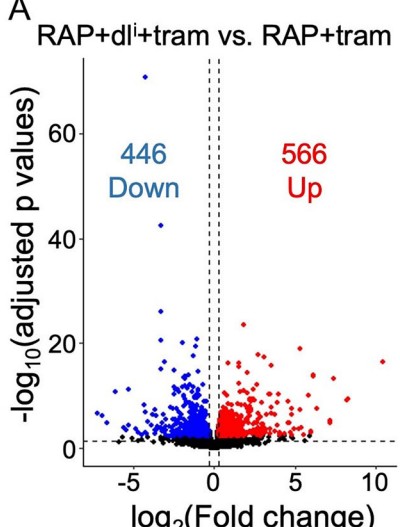

A

RAP+dl$^i$+tram vs. RAP+tram

446
Down

566
Up

log$_2$(Fold change)

-log$_{10}$(adjusted p values)

B

| Pathway | Genes |
|---|---|
| Toll | *pgrp-sa, pgrp-sd, grass, drs, atta, wntd* |
| Imd | *pgrp-lb, dptb, pirk* |
| Jak/Stat | *upd1, upd2, upd3, socs36e, dome* |
| Jnk | *kay, jra, puc, mmp1* |
| Glucuronidation | *sgl, pgd, rpi, zw* |

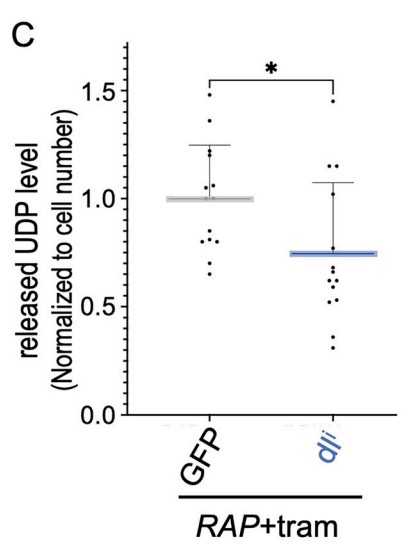

C

released UDP level
(Normalized to cell number)

GFP    dl$^i$

RAP+tram

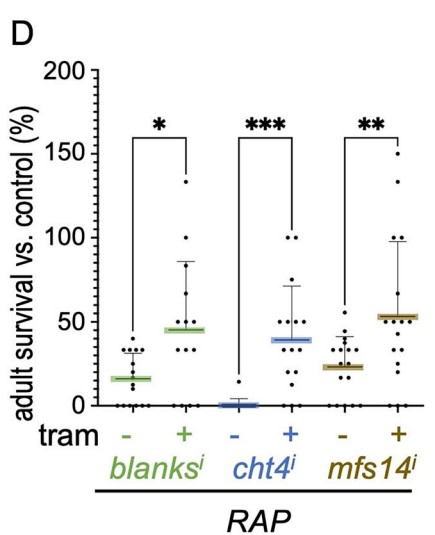

D

adult survival vs. control (%)

tram   -   +   -   +   -   +

*blanks$^i$*   *cht4$^i$*   *mfs14$^i$*

RAP

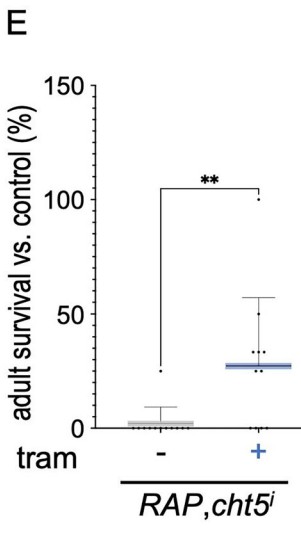

E

adult survival vs. control (%)

tram   -   +

*RAP,cht5$^i$*

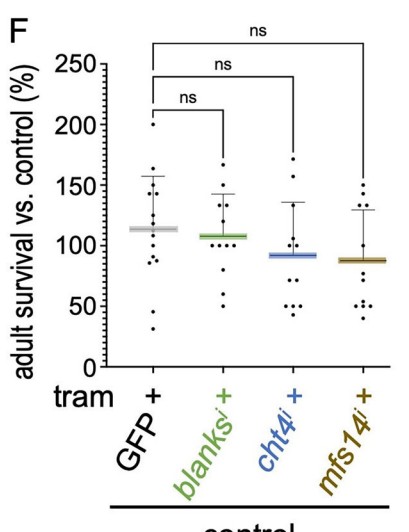

F

adult survival vs. control (%)

ns

ns

ns

tram   +   +   +   +

GFP   *blanks$^i$*   *cht4$^i$*   *mfs14$^i$*

control

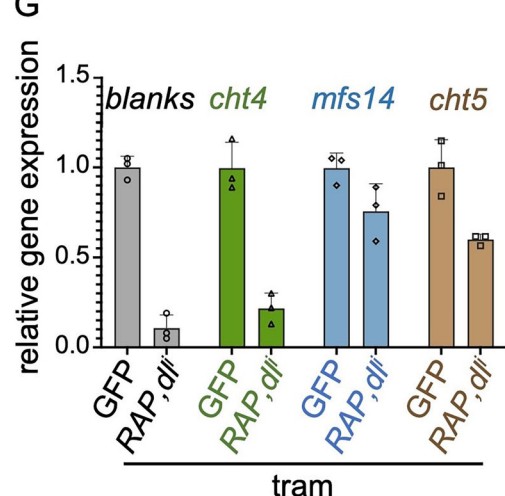

G

relative gene expression

*blanks*   *cht4*   *mfs14*   *cht5*

GFP   RAP,dl$^i$   GFP   RAP,dl$^i$   GFP   RAP,dl$^i$   GFP   RAP,dl$^i$

tram

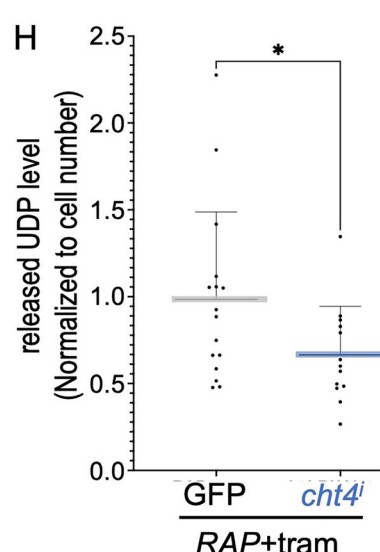

H

released UDP level
(Normalized to cell number)

GFP    cht4$^i$

RAP+tram

**Figure 3.  Canonical NF-κB activity affects trametinib response by regulating its glucuronidation.**

(A) Volcano plot showing log2 fold change (x-axis) and −log10 adjusted $p$ values (y-axis) of genes differentially expressed between *RAP;dl*[i] and *RAP* in the presence of trametinib (1 μM). (B) Examples of significantly altered pathways. (C, H) The levels of trametinib glucuronidation measured by released UDP level in *RAP;GFP* (tram, $n = 14$), *RAP;dl*[i] (tram, $n = 14$) (C) or *RAP;GFP* (tram, $n = 16$), *RAP;cht4*[i] (tram, $n = 13$) (H). (D–F) Percent survival of transgenic flies to adulthood relative to control flies was quantified in the presence or absence of trametinib (1 μM). (D) *RAP;blanks*[i] (DMSO, $n = 16$; tram, $n = 14$), *RAP;cht4*[i] (DMSO, $n = 18$; tram, $n = 16$), *RAP;mfs14*[i] (DMSO, $n = 17$; tram, $n = 17$); (E) *RAP;cht5*[i] (DMSO, $n = 16$; tram, $n = 13$); (F) Control (tram, $n = 15$), *blanks*[i] (tram, $n = 12$), *cht4*[i] (tram, $n = 12$), *mfs14*[i] (tram, $n = 12$). (G) Gene expression (*blanks, cht4, cht5* and *mfs14*) was detected by RNA sequencing in *RAP;GFP* ($n = 3$) and *RAP;dl*[i] ($n = 3$). (A, C, G, H) The experiments were conducted at 27 °C. (D–F) The experiment was conducted at 29 °C. The statistical tests used to calculate the $P$ value are as follows: (A) Wald test; (C, F, H) Mann–Whitney test; (D, E) one-way ANOVA; NS $P(>0.12)$, *$P(0.033)$, **$P(0.002)$ and ***$P(<0.001)$. All statistical data are summarised in Table EV1. The error bar is a standard deviation (SD), with each point representing biological replicates and numbers ($n$), including three technical replicates. Source data are available online for this figure.

particular, Wnt pathway inhibitors PNU-74654 (Trosset et al, 2006) and LF3 (Fang et al, 2016)—which act by suppressing the interaction between β-Catenin and TCF—demonstrated strong efficacy in reducing *RAP* tumours when paired with trametinib (Fig. 4B,C). This suppression led to increased animal survival in the presence of trametinib, without impacting survival in the absence of trametinib or in control animals (Fig. 4D). Consistent with our results, combining trametinib plus PNU-74654 strongly suppressed canonical NF-κB activity in *RAP* tumours (Fig. EV4E).

Regarding tumour progression, PNU-74654 plus trametinib suppressed tumour overgrowth in the HPZ, while PNU-74654 alone did not affect tumour overgrowth (compare Figs. 1H to 4F,G, quantified in 4E). These data indicate that targeting Wnt activity pharmacologically is effective at reducing trametinib resistance in *RAP* tumours.

## Combining trametinib and PNU-74654 suppressed tumour progression in genetically complex 'avatar' tumours

In previous work (Bangi et al, 2016), we found that fly CRC models targeting three to four genes responded poorly to trametinib as a single agent, requiring drug combinations for efficacy. We therefore assessed whether combining trametinib plus PNU-74654 could effectively suppress tumour progression in still more genetically complex CRC lines. We tested seven 'patient-specific fly avatar' lines, each targeting 6-10 genes to more fully model the mutation profile of individual CRC patients (Fig. EV4F). Each exhibited minimal response to trametinib alone (Fig. 5). In contrast, five CRC avatar lines designed as part of a clinical trial (Bangi et al, 2019) responded significantly to oral trametinib plus PNU-74654 (Fig. 5A–E), while two additional lines developed from the TCGA cancer database (Network, 2013) did not significantly respond to the cocktail (Fig. 5F).

## Mediators of trametinib/PNU-74654 efficacy in CRC tumours

To identify factors that mediate the efficacy of trametinib plus PNU-74654 on *RAP* flies, we performed a genetic screen (Figs. 6A and EV5A). We identified five genes that, when targeted for knockdown by RNA-interference (RNAi), enhanced adult eclosion of *RAP* flies in the presence of trametinib plus PNU-74654 (or LF3): *brahma* (*brm*, orthologue of human *SMARCA2* and *SMARCA4*), *shotgun* (*shg, CDH1*), *archipelago* (*ago, FBXW7*), *rhoGAPp190* (*rhoGAPp190, ARHGAP5, ARHGAP35*) and *upf1 RNA helicase* (*upf1, UPF1*) (Figs. 6A and EV5B). Importantly, none

of these five gene knockdowns impacted survival of control animals or untreated *RAP* flies (Figs. 6B,C and EV5C). These data suggest that *brm, shg, ago, rhoGAPp190* and *upf1* help mediate the efficacy of trametinib plus a WNT pathway inhibitor in CRC tumours. Testing single drugs with each of the five loci, we found that *brm*[i] and *ago*[i] significantly increased the sensitivity of *RAP* tumours to trametinib (Fig. 6E, schematised in 6D); *rhoGAPp190*[i] sensitised *RAP* tumours to PNU-74654 (Fig. 6F, schematised in 6D).

As noted above, two of seven tested 'patient-specific fly avatar' lines, *RAPp1* and *RAPp2*, were resistant to combined trametinib plus PNU-74654 (Fig. 5F). Knockdown of *brm, shg, ago, rhoGAPp190* and *upf1* each significantly enhanced tumour sensitivity to trametinib plus PNU-74654 in both multigenic tumours, with the exception that *shg* had only weakly improved *RAPp1* drug response (Fig. 6G,H). Finally, we note human orthologues of these five loci are altered in a subset of human CRC patients (Fig. EV5D), suggesting they could serve as biomarkers for patients that would be especially sensitive to trametinib plus PNU-74654.

## Elevated WNT plus KRAS is associated with increased canonical NF-κB signalling in human CRC tumour samples

To help assess whether our *Drosophila* data were relevant to human CRC, we examined human KRAS-mutant CRC tissue sections to determine whether high WNT activity is associated with high NF-κB activity. We used IKK isoforms to identify canonical vs. non-canonical NF-κB signalling. IKKβ serves as the primary catalytic subunit of IKK, activating canonical NF-κB signalling by proin-flammatory cytokines such as TNFα, IL-1 and LPS. IKKα activates non-canonical NF-κB signalling activated by other members of the TNFR superfamily (Häcker and Karin, 2006; Scheidereit, 2006): for example, IKKα is phosphorylated by NIK at Ser-176 to promote release of the non-canonical NF-κB factor RelB-p52 into the nucleus to activate target genes (Ling et al, 1998).

WNT activity was assessed in human CRC patient samples by anti-β-catenin antibody (Fig. EV6A,B). We observed that expression of the IKKβ protein was significantly higher in CRC patient samples with both elevated WNT activity plus oncogenic KRAS when compared to those with only elevated WNT activity *or* KRAS mutations (Fig. 7A–D, quantified in 7I). In contrast, we found no significant differences in the levels of IKKα or Ser-176 phosphory-lated IKKα in samples with (i) high WNT/oncogenic KRAS samples vs. (ii) high WNT activity or oncogenic KRAS (Fig. 7E–H, quantified in 7J; Fig. EV6C–F, quantified in EV6G). That is, CRC tumours with oncogenic KRAS plus high WNT activity are

## A

| Compounds | Targets | Rescue rate (Combinations/Trametinib) |
|---|---|---|
| PNU-74654 | Wnt/β-catenin | 19.58% |
| LF3 | Wnt/β-catenin | 15.45% |
| iCRT3 | Wnt/β-catenin | 10.76% |
| IWP-o1 | porcupine | 8.23% |
| XAV-939 | tankyrase1/2 | 8.37% |
| Capmatinib | Wnt/β-catenin and EMT signaling | 6.62% |
| IWR-1-endo | Wnt | 1.4% |
| Wnt-C59 | PORCN | - |

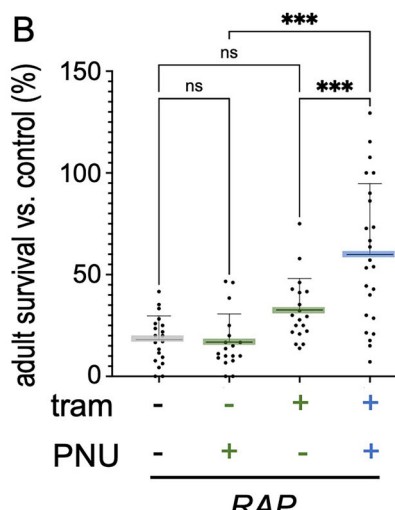

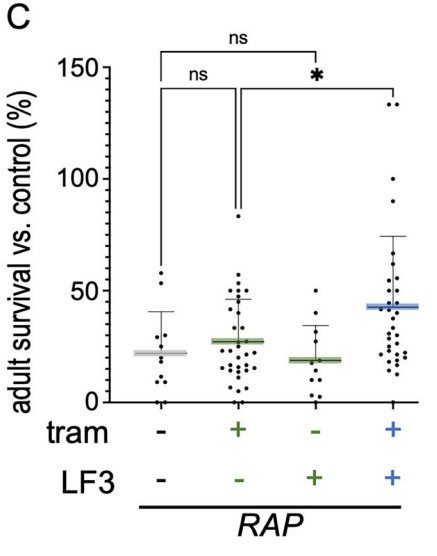

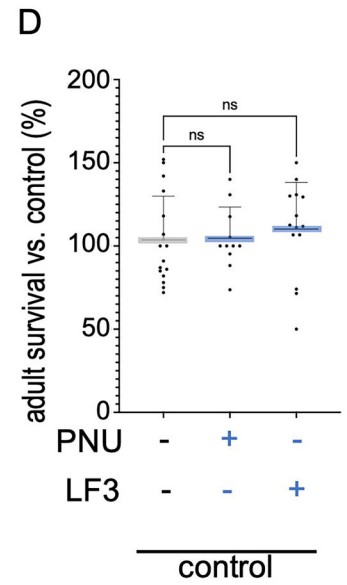

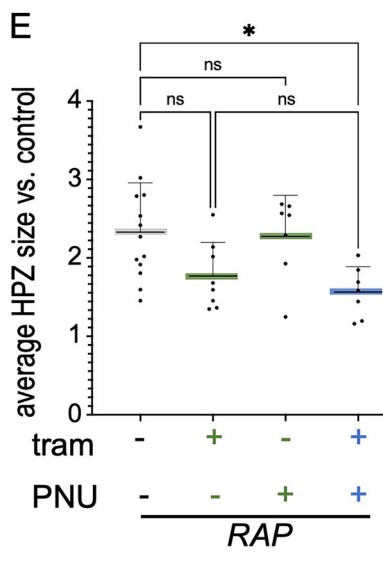

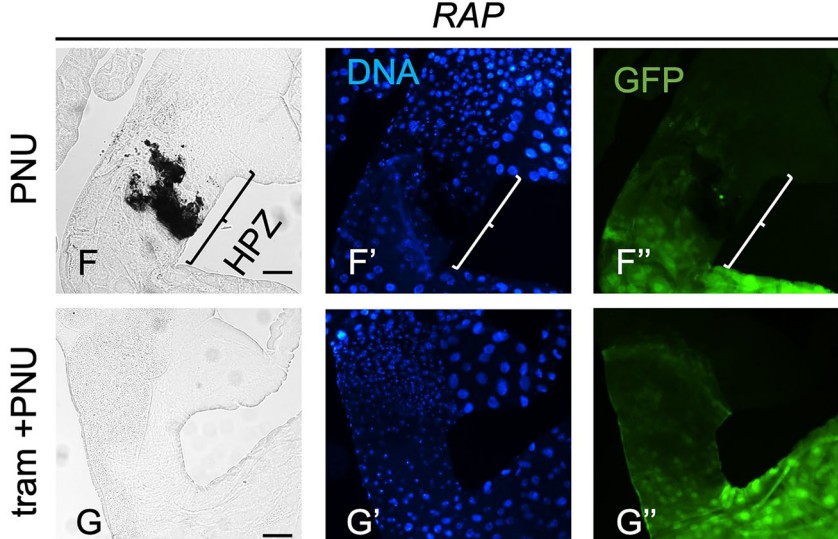

**Figure 4.   PNU-74654 and LF3 suppressed trametinib resistance in *RAP* tumours.**

(A) Summary of the rescue rate of trametinib and WNT inhibitor drug combinations in *RAP* hindgut tumours. (B–D) Percent survival of *RAP* flies to adulthood relative to control flies was quantified in the presence or absence of trametinib (1 μM), PNU-74654 (1 μM) or LF3 (10 μM). *RAP* (DMSO, n = 20; tram, n = 19; PNU, n = 18; tram +PNU, n = 24) (B); *RAP* (DMSO, n = 12; tram, n = 34; LF3, n = 12; tram + LF3 n = 33) (C); Control (DMSO, n = 17; tram+PNU, n = 11; tram + LF3 n = 14) (D). (E) The average hindgut proliferation zone (HPZ) size was measured by Fiji ImageJ and quantified as relative size to wild-type (WT) hindguts, *RAP* (DMSO, n = 13; tram, n = 8; PNU, n = 7; tram + PNU, n = 7). (F, G) Images of the digestive tract of third instar larvae in the presence or absence of trametinib (1 μM) or PNU-74654 (1 μM), Scale bar 100 μm. Scale bar 200 μm. (A–G) The experiment was conducted at 27 °C. The statistical tests used to calculate the P value are as follows: (B–E) one-way ANOVA; NS P(>0.12), *P(0.033), **P(0.002) and ***P(<0.001). All statistical data are summarised in Table EV1. The error bar is a standard deviation (SD), with each point representing biological replicates and numbers (n), including three technical replicates. Source data are available online for this figure.

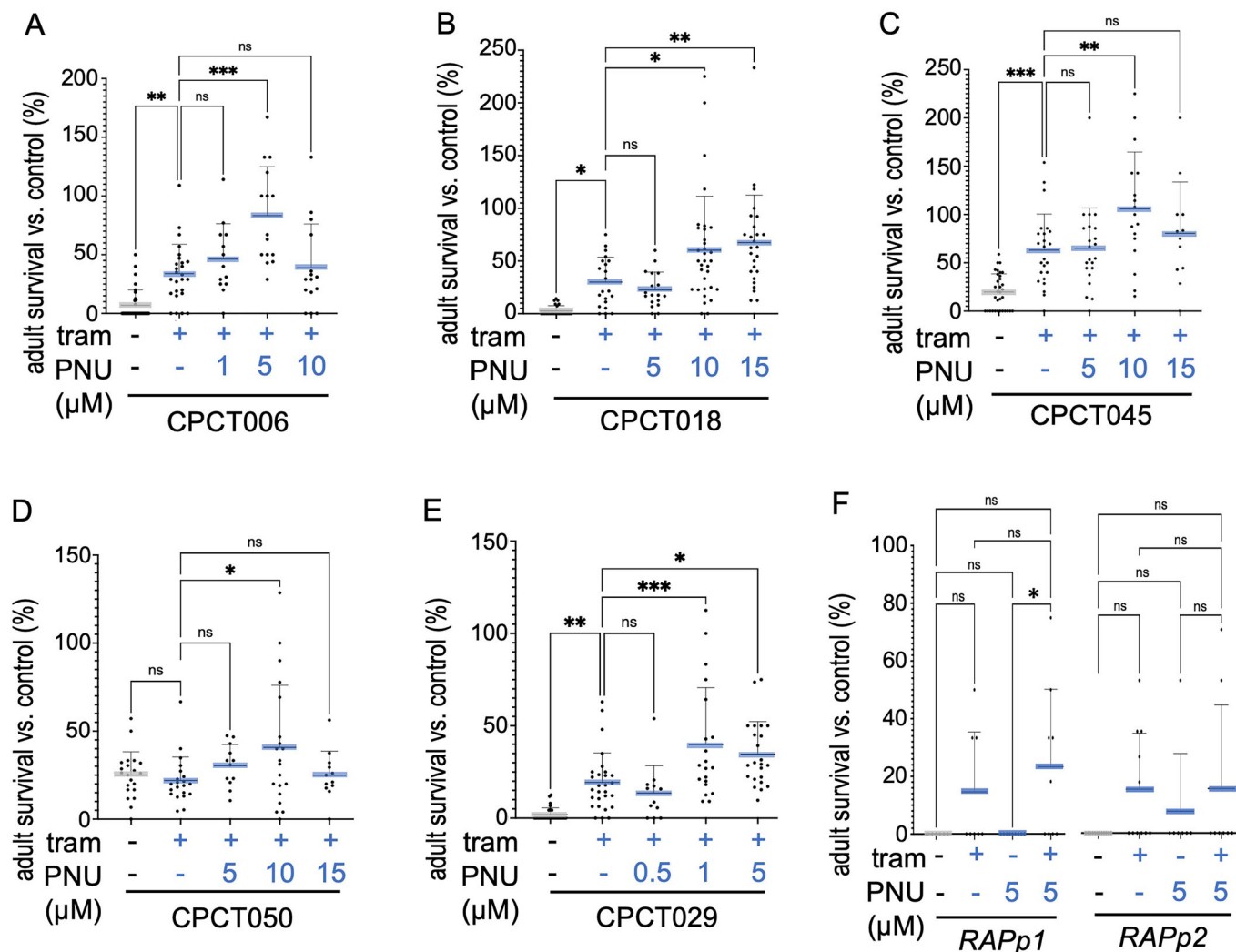

**Figure 5.   Combination of trametinib and PNU-74654 suppressed tumour progression in various genetically complex tumours.**

(A–F) Percent survival of transgenic patient-specific avatar fly lines to adulthood relative to control flies was quantified in the presence or absence of trametinib (1 μM) or PNU-74654. (A) *CPCT006* (DMSO, n = 33; tram, n = 26; tram + 1 μM PNU, n = 13; tram + 5 μM PNU, n = 15; tram + 10 μM PNU, n = 15); (B) *CPCT018* (DMSO, n = 24; tram, n = 21; tram + 0.5 μM PNU, n = 18; tram + 1 μM PNU, n = 33; tram + 5 μM PNU, n = 26); (C) *CPCT045* (DMSO, n = 33; tram, n = 24; tram + 5 μM PNU, n = 22; tram + 10 μM PNU, n = 17; tram + 15 μM PNU, n = 12); (D) *CPCT050* (DMSO, n = 21; tram, n = 21; tram + 5 μM PNU, n = 12; tram + 10 μM PNU, n = 20; tram + 15 μM PNU, n = 12); (E) *CPCT029* (DMSO, n = 28; tram, n = 28; tram + 0.5 μM PNU, n = 12; tram + 1 μM PNU, n = 20; tram + 5 μM PNU, n = 23). (F) trametinib and/or PNU-74654 (5 μM) exhibited poor rescue of *RAPp1* and *RAPp2*. *RAPp1* (DMSO, n = 7; tram, n = 8; PNU, n = 9; tram + PNU, n = 9); *RAPp2* (DMSO, n = 12; tram, n = 11; PNU, n = 7; tram + PNU, n = 8). (A) The experiment was conducted at 25 °C. (B) and (D–F) The experiments were conducted at 28 °C. (C) The experiment was conducted at 27 °C. The statistical tests used to calculate the P value are as follows: (A–F) one-way ANOVA; NS P(>0.12), *P(0.033), **P(0.002) and ***P(<0.001). All statistical data are summarised in Table EV1. The error bar is a standard deviation (SD), with each point representing biological replicates and numbers (n), including three technical replicates. Source data are available online for this figure.

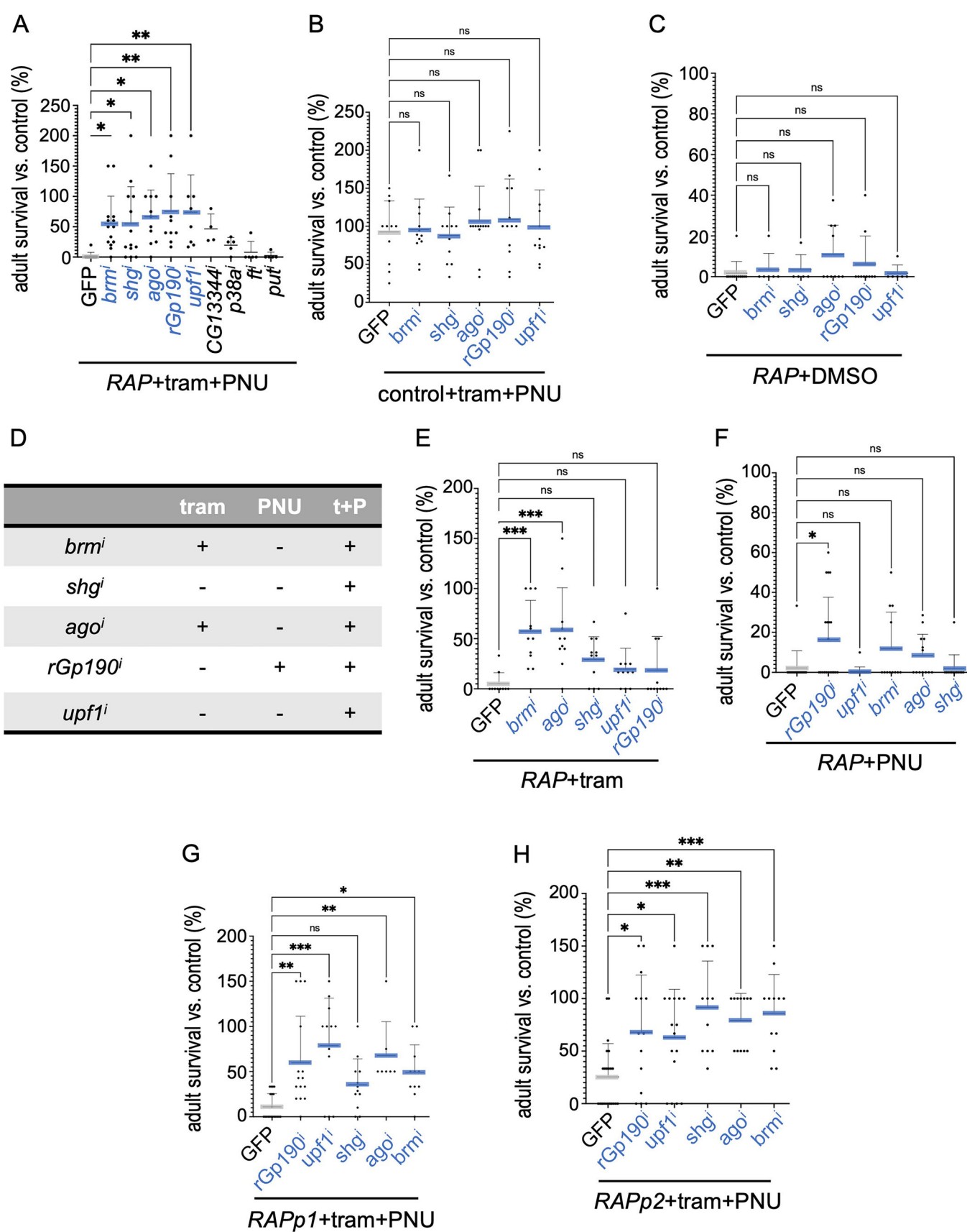

**Figure 6.  Regulators of the combination of trametinib and PNU-74654 in CRC tumours.**

(A–C) Survival of transgenic flies to adulthood relative to control flies was quantified in the presence or absence of trametinib (1 μM) or PNU-74654 (1 μM). (A) Knockdown of *brm*, *shg*, *ago*, *rhoGAPp190*, or *upf1* improved the survival to adulthood of *RAP* flies in the presence of trametinib plus PNU-74654. *RAP;GFP* (tram + PNU, n = 11), *RAP;brm* (tram+PNU, n = 14), *RAP;ago* (tram+PNU, n = 11), *RAP;shg* (tram + PNU, n = 13), *RAP;upf1* (tram + PNU, n = 8), *RAP;rhoGAPp190* (tram + PNU, n = 10). (B) *GFP* (tram + PNU, n = 11), *brm* (tram + PNU, n = 12), *ago* (tram + PNU, n = 14), *shg* (tram + PNU, n = 11), *upf1* (tram + PNU, n = 12), *rhoGAPp190* (tram + PNU, n = 13). (C) *RAP;GFP* (DMSO, n = 12), *RAP;brm* (DMSO, n = 6), *RAP;ago* (DMSO n = 10), *RAP;shg* (DMSO, n = 5), *RAP;upf1* (DMSO, n = 6), *RAP;rhoGAPp190* (DMSO n = 10). (D) Summary of five loci found to impact *RAP* response to trametinib and/or PNU-74654. (E–H) Survival of transgenic flies to adulthood relative to control flies was quantified in the presence or absence of trametinib (1 μM) or PNU-74654 (1 μM except where noted). (E) RNAi-mediated knockdown of *brm* or *ago* improved the survival of *RAP* flies treated with trametinib. *RAP;GFP* (tram, n = 10), *RAP;brm* (tram, n = 11), *RAP;ago* (tram, n = 11), *RAP;shg* (tram, n = 11), *RAP;upf1* (tram, n = 11), *RAP;rhoGAPp190* (tram, n = 11). (F) *rhoGAPp190* knockdown improved the survival of *RAP* flies treated with PNU-74654. *RAP;GFP* (PNU, n = 15), *RAP;brm* (PNU, n = 12), *RAP;ago* (PNU, n = 13), *RAP;shg* (PNU, n = 13), *RAP;upf1* (PNU, n = 20), *RAP;rhoGAPp190* (PNU, n = 20). (G, H) Knockdown of *brm*, *shg*, *ago*, *rhoGAPp190*, or *upf1* rescued more genetically complex avatar lines *RAPp1* and *RAPp2* when treated with trametinib plus PNU-74654 (5 μM); note *shg* rescue of *RAPp1* did not rise to the level of statistical significance. *RAPp1;GFP* (tram+PNU, n = 20), *RAPp1;brm* (tram+PNU, n = 11), *RAPp1;ago* (tram + PNU, n = 7), *RAPp1;shg* (tram + PNU, n = 12), *RAPp1;upf1* (tram + PNU, n = 12), *RAPp1;rhoGAPp190* (tram + PNU, n = 15) (G); *RAPp2;GFP* (tram + PNU, n = 22), *RAPp2;brm* (tram + PNU, n = 12), *RAPp2;ago* (tram + PNU, n = 12), *RAPp2;shg* (tram + PNU, n = 11), *RAPp2;upf1* (tram + PNU, n = 16), *RAPp2;rhoGAPp190* (tram + PNU, n = 14) (H). The statistical tests used to calculate the *P* value are as follows: (A–C and E–H) one-way ANOVA; NS *P*(>0.12), *\**P*(0.033), *\*\**P*(0.002) and *\*\*\**P*(<0.001). All statistical data are summarised in Table EV1. The error bar is a standard deviation (SD), with each point representing biological replicates and numbers (*n*), including three technical replicates. Source data are available online for this figure.

associated with significantly elevated canonical NF-κB signalling, consistent with our Drosophila data. Of note, we observed upregulation of glucuronidation in the presence of trametinib in human T84 colon cancer cells; JSH-23 administration led to a trend of enhancing trametinib efficacy that only rose to the level of significance in 1/3 experiments (Figs. 8A,B and EV6H). In contrast, SW620 colon cancer cells were significantly more sensitive to treatment with JSH-23-plus-trametinib (Figs. 8C and EV6I). Both cell lines contain KRAS and APC mutations, suggesting that additional genetic mutations can influence drug response.

# Discussion

Colorectal tumours that contain oncogenic RAS isoforms have proven resistant to most targeted therapies. In this study, we conduct an in-depth analysis of 'RAP' tumours that contain the three genes most commonly mutated in human CRC: *RAS* (typically *KRAS*), *APC*, and *P53*. We found that pairing oncogenic Ras^G12V with elevated Wnt activity led to upregulation of canonical NF-κB signalling, both in *Drosophila* hindgut tumours and human tumours. This emergent canonical NF-κB activity has consequences: it reduced the efficacy of the MEK inhibitor trametinib in *RAP* tumours at least in part by increasing glucuronidation of the drug (Fig. 8D). This resistance to trametinib was strongly reversed when WNT inhibitors were included. Oncogenic RAS isoforms are present in approximately half of all CRC tumours, and our work provides new pathways towards benefiting these patients.

Most WNT pathway inhibitors currently undergoing clinical trials act by promoting degradation of β-catenin, including inhibitors of PORCN (e.g. ETC-1922159, WNT974 and XNW7201) and Frizzled receptors (e.g. vanticumab, ipafricept) (Neiheisel et al, 2022). However, our data suggest that the class of WNT inhibitors that disrupt the interaction between β-catenin and TCF—including PNU-74654 and LF3—are particularly effective when used in combination with trametinib for treating CRC. Further, this drug combination proved effective even in more genetically complex CRC avatar lines that were especially resistant to a broad palette of drugs, including trametinib. To extend our drug resistance work, we identified five genes that further enhanced

the efficacy of trametinib/PNU-74654: *brm*, *shg*, *ago*, *rhoGAPp190* and *upf1*. These genes are mutated in a subset of patients, identifying a cohort that may prove especially responsive to MEK/WNT inhibitor drug combinations. Two genes—*brm* and *ago*—enhanced the efficacy of trametinib alone, identifying a candidate biomarker for trametinib response. That is, our work identifies a path to matching drugs to specific subsets of RAS-mutant CRC patients.

The NF-κB pathway has been widely linked to cancer, including impacting drug resistance by regulating the survival of cancer cells. For example, NF-κB activity is reported to inhibit the response of HTC15 human colon cancer cells to daunomycin by controlling drug uptake (Bentires-Alj et al, 2003). NF-κB activity is also linked to sorafenib resistance in CD13^+ hepatocellular carcinoma cell lines by controlling genes that regulate cell cycle and apoptosis (Hu et al, 2020). Pharmacologically blocking the NF-κB pathway sensitises tumour cells to doxorubicin in Dll1+ mouse breast cancer cells by promoting cell death (Kumar et al, 2021). In this whole animal study, we demonstrate that canonical NF-κB-mediated drug resistance—through upregulation of glucuronidation pathway activity—is an emergent property of CRC tumours that combine high WNT activity with oncogenic Ras. This may have therapeutic implications, as we demonstrate.

We previously showed a role for TNF signalling in regulating tumour progression in a Ras-dependent *Drosophila* cancer model (Cordero et al, 2010) and, indeed, removing just one genomic copy of the TNF pathway mediator *dl* was sufficient to significantly suppress tumour-induced animal lethality. However, pharmacological inhibition of canonical NF-κB pathway activity by JSH-23 did not strongly suppress *RAP* tumour-induced lethality, while the NF-κB signalling inhibitor QNZ (EVP4593) enhanced trametinib resistance in *RAP* tumours. QNZ (EVP4593) targets TNFα production, suggesting that systemic TNFα production plays a role in inhibiting tumour progression in the presence of trametinib. Indeed, it has been reported that TNFα renders tumour vessels more permeable, facilitating the delivery of anticancer drug agents to solid tumours (Seynhaeve et al, 2007). Together, pharmacological inhibition—through canonical NF-κB pathway activity inhibitor JSH-23 or signalling inhibitor QNZ (EVP4593)—did not significantly suppress RAP tumour-induced lethality below

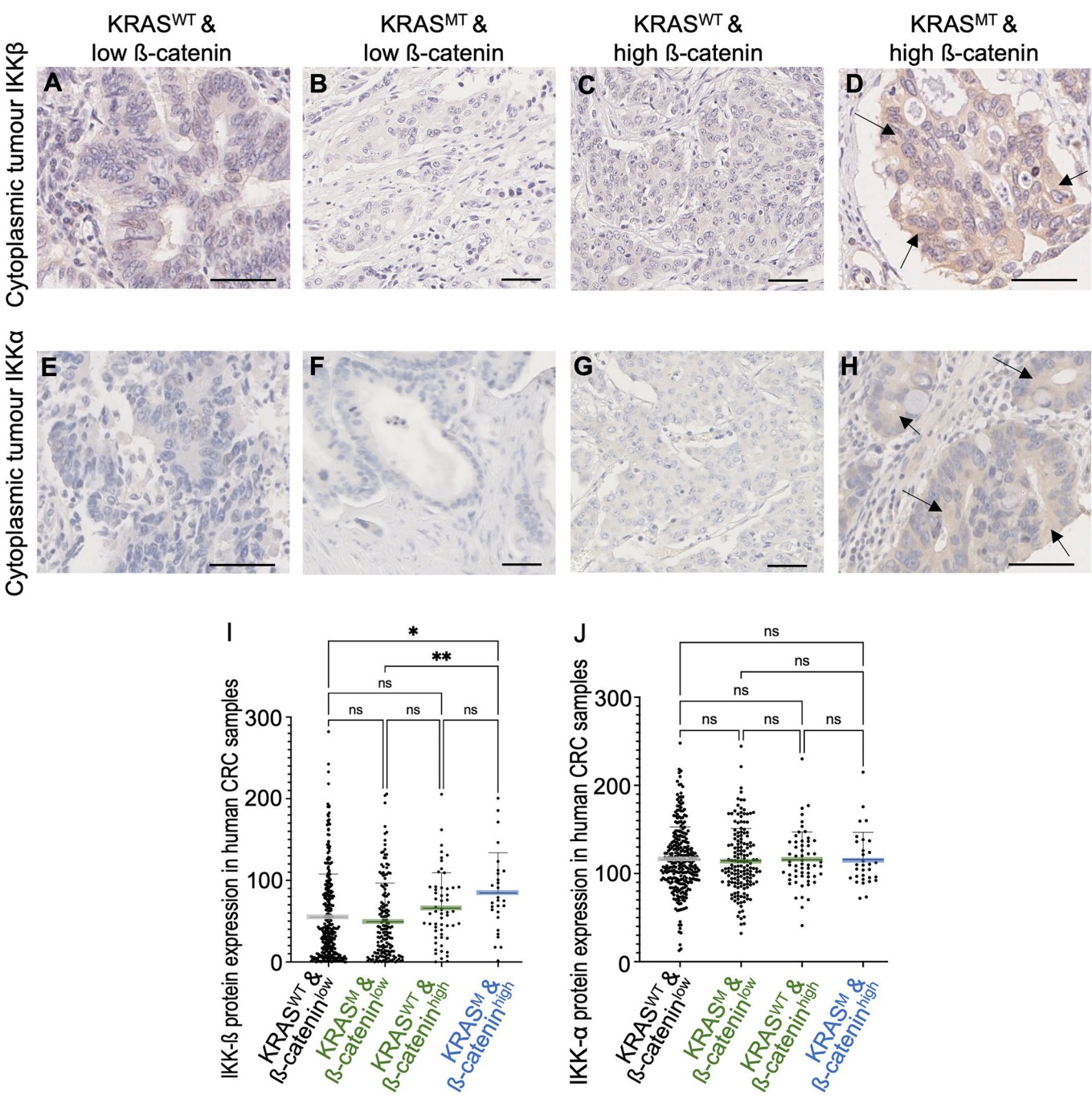

**Figure 7. Co-activation of WNT and KRAS was associated with an upregulation of canonical NF-κB activity in human CRC.**

(A–H) Representative immunohistochemical (IHC) staining for IKKβ and IKKα in stage 2–3 colorectal cancer patient samples. (A, E) $KRAS^{WT}$ (wild-type) plus low β-catenin expression CRC samples; (B, F) $KRAS^{MT}$ (mutation in position G12/G13) plus low β-catenin expression CRC samples; (C, G) $KRAS^{WT}$ plus high β-catenin expression CRC samples; (D, H) $KRAS^{MT}$ plus high β-catenin expression CRC samples. The black arrow highlights the high signal area. Scale bar 50 μm. (I, J) The graph shows the mean of expression of IKKβ (I) or IKKα (J) in each different mutated human CRC, determined by IHC intensity values. Patients were grouped into four categories based on KRAS status and β-catenin expression. (I) Both $KRAS^{WT}$ and β-catenin low ($n = 311$); $KRAS^{M}$ and β-catenin low ($n = 164$); $KRAS^{WT}$ and β-catenin high ($n = 56$); Both $KRAS^{M}$ and β-catenin high ($n = 28$). (J) Both $KRAS^{WT}$ and β-catenin low ($n = 291$); $KRAS^{M}$ and β-catenin low ($n = 158$); $KRAS^{WT}$ and β-catenin high ($n = 57$); Both $KRAS^{M}$ and β-catenin high ($n = 31$). Each dot displays an individual sample. The statistical tests used to calculate the $P$ value are as follows: (I, J) one-way ANOVA; NS $P(>0.12)$, *$P(0.033)$, **$P(0.002)$ and ***$P(<0.001)$. All statistical data are summarised in Table EV1. The error bar is a standard deviation (SD), with each point representing biological replicates. Source data are available online for this figure.

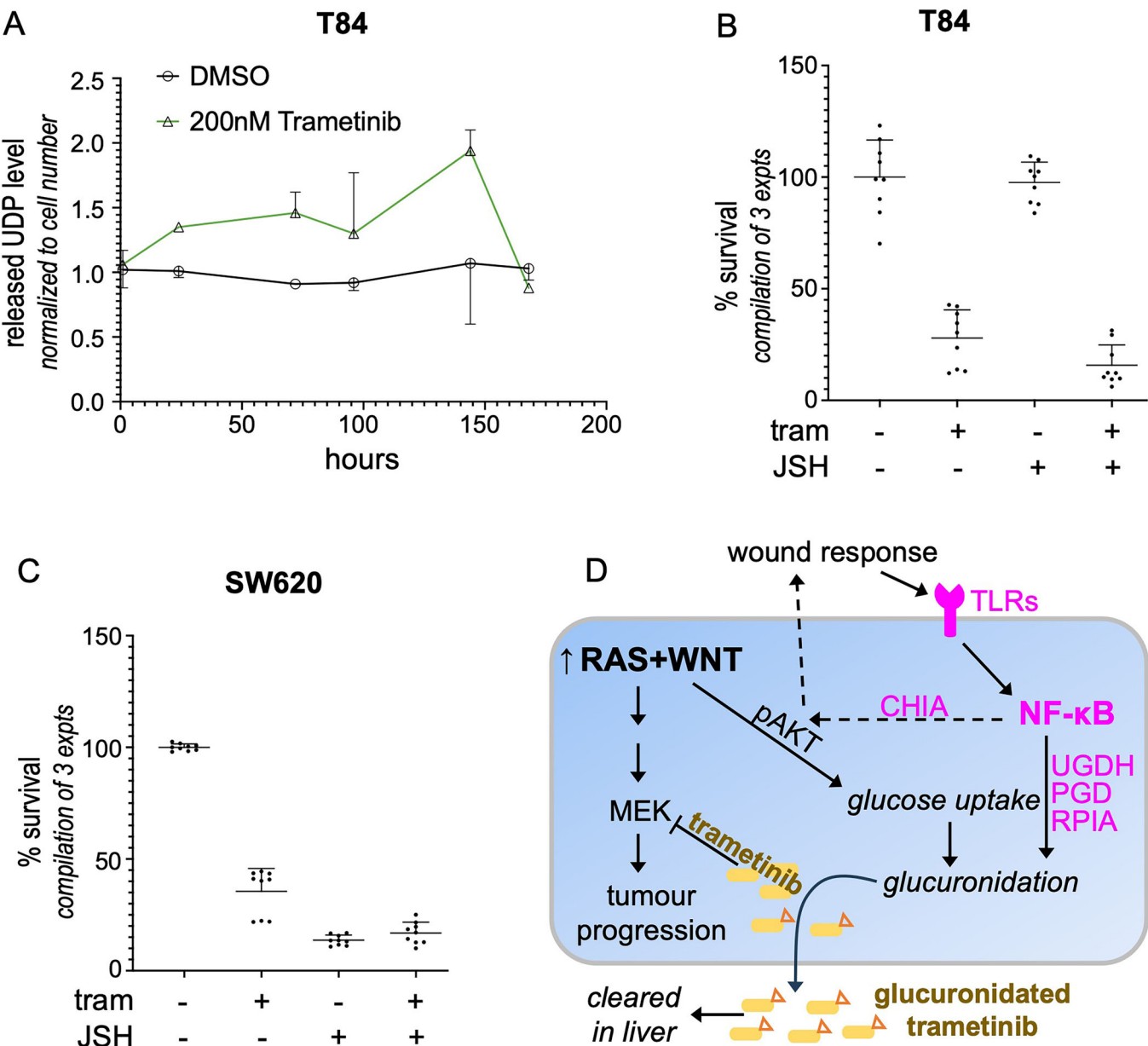

**Figure 8. The efficacy of trametinib or JSH-23 on human colon cancer cells.**

(A) The levels of trametinib glucuronidation measured by released UDP level in T84 colon cancer cells, $n = 9$. (B, C) Cell viability assay for T84 colon cancer cells (B) or SW620 (C) treatment with 0.1% DMSO, 50 nM trametinib or 10 μM JSH-23, $n = 9$. The error bar is a standard deviation (SD), with each point representing biological replicates and numbers (n), including three technical replicates. (D) Schematic summary. We previously reported that pairing activated RAS and WNT activities leads to increased glucose flux into cells in a PI3K/AKT pathway-dependent manner, leading to elevated glucuronidation and elimination of the potent MEK inhibitor trametinib in cancer cells (Cong et al, 2025). Here, we provide evidence that elevated WNT signalling leads to a wound-like response in RAS cancer cells that induced upregulation of canonical NF-κB activity by Toll-like receptors (TLRs). This upregulation enhances glucuronidation pathway activity by increasing gene expression of related enzymes including controlling the CHIA/AKT glucose uptake axis. Source data are available online for this figure.

toxic levels. This suggests that broad targeting of NF-κB may be problematic from a whole-body standpoint, not surprising given the large number of processes controlled by NF-κB activity.

Genetic complexity is a common clinical feature of tumours. Here we demonstrate that a common version of this complexity linked to aggressive CRC disease—combining alterations in RAS, APC, and P53—is sufficient to direct overgrowth of the hindgut proliferative zone (HPZ) and promote emergent drug resistance. Currently, patient tumours with these three altered genes have few second-line therapeutic options, as RAS pathway inhibitors have failed to provide durable responses. Gaining a deeper understanding of how this combination directs drug resistance, through factors such as NF-κB, will provide new candidate avenues towards therapeutics.

# Methods

## Reagents and tools table

| Reagent/resource | Reference or source | Identifier or catalogue number |
|---|---|---|
| **Experimental models** | | |
| byn-gal4 | V. Hartenstein | N/A |
| UAS-Ras[G12V] | G. Halder | N/A |
| tub-gal80[TS] | BDSC | 7017 |
| w[1118] | BDSC | 3605 |
| UAS-mCD8-GFP | BDSC | 5137 |
| UAS-dl[i] | BDSC | 36650 |
| UAS-dif[i] | BDSC | 30513 |
| UAS-cact[i] | BDSC | 37484 |
| UAS-Arm[S10] | BDSC | 4782 |
| UAS-Toll1[i] | BDSC | 35628 |
| UAS-Toll3[i] | BDSC | 28526 |
| UAS-Toll4[i] | BDSC | 28543 |
| UAS-Toll5[i] | BDSC | 29533 |
| UAS-Toll6[i] | BDSC | 56048 |
| UAS-Toll7[i] | BDSC | 30488 |
| UAS-Toll8[i] | BDSC | 28519 |
| UAS-Toll9[i] | BDSC | 34853 |
| UAS-brm[i] | BDSC | 35211 |
| UAS-shg[i] | BDSC | 38207 |
| UAS-ago[i] | BDSC | 34802 |
| UAS-rhoGAPp190[i] | BDSC | 43987 |
| UAS-upf1[i] | BDSC | 64519 |
| UAS-CG13344[i] | BDSC | 41831 |
| UAS-p38a[i] | BDSC | 35244 |
| UAS-ft[i] | BDSC | 34970 |
| UAS-put[i] | BDSC | 39025 |
| UAS-dnapol-eta[i] | BDSC | 33410 |
| UAS-lrp1[i] | BDSC | 44579 |
| UAS-tefu[i] | BDSC | 44073 |
| UAS-nej[i] | BDSC | 37489 |
| UAS-nos[i] | BDSC | 33973 |
| UAS-pc[i] | BDSC | 36070 |
| UAS-rad51c[i] | BDSC | 67355 |
| dl[1] | BDSC | 3236 |
| UAS-blanks[i] | BDSC | 33667 |
| UAS-cht4[i] | BDSC | 65001 |
| UAS-mfs14[i] | BDSC | 33999 |
| UAS-cht5[i] | BDSC | 57512 |
| UAS-ref(2)p[i] | BDSC | 36111 |
| UAS-CG32302[i] | BDSC | 67239 |
| UAS-CG17104[i] | BDSC | 42925 |
| UAS-mec2[i] | BDSC | 61259 |
| UAS-CG15739[i] | BDSC | 57216 |
| UAS-ag5r[i] | BDSC | 67225 |

| Reagent/resource | Reference or source | Identifier or catalogue number |
|---|---|---|
| UAS-arc1[i] | BDSC | 25954 |
| UAS-CG2065[i] | BDSC | 55283 |
| UAS-CG10182[i] | BDSC | 61954 |
| UAS-CG18473[i] | BDSC | 57524 |
| UAS-cdc23[i] | BDSC | 61982 |
| UAS-rpt3r[i] | BDSC | 58140 |
| UAS-ter94[i] | BDSC | 35608 |
| UAS-prosalpha4[i] | BDSC | 65161 |
| UAS-alh[i] | BDSC | 39057 |
| UAS-punch[i] | BDSC | 41998 |
| UAS-CG4502[i] | BDSC | 35489 |
| UAS-CG12493[i] | BDSC | 42791 |
| UAS-vis[i] | BDSC | 35738 |
| UAS-rpt4[i] | BDSC | 32874 |
| UAS-rpn3[i] | BDSC | 34561 |
| UAS-cyp4p1[i] | BDSC | 67349 |
| UAS-hsp23[i] | BDSC | 82961 |
| UAS-mal-a6[i] | BDSC | 60398 |
| UAS-fng[i] | BDSC | 25947 |
| UAS-CG30427[i] | BDSC | 58271 |
| UAS-prosbeta3[i] | BDSC | 34868 |
| UAS-idh[i] | BDSC | 41708 |
| UAS-CG8036[i] | BDSC | 60371 |
| UAS-nd-pdsw[i] | BDSC | 29592 |
| UAS-cda4[i] | BDSC | 65909 |
| UAS-muc[i] | BDSC | 44439 |
| UAS-CG4459[i] | BDSC | 61228 |
| UAS-cox5a[i] | BDSC | 58282 |
| UAS-CG32564[i] | BDSC | 58342 |
| UAS-nd-b22[i] | BDSC | 65011 |
| UAS-cox7a[i] | BDSC | 57572 |
| UAS-rel[i] | BDSC | 33661 |
| UAS-domeHMJ21208[i] | BDSC | 53890 |
| UAS-domeHMS10293[i] | BDSC | 34618 |
| UAS-bsk[i] | BDSC | 36643 |
| UAS-CG9360[i] | VDRC | v13189 |
| UAS-CG1698[i] | VDRC | v101947 |
| UAS-CG32365[i] | VDRC | v104119 |
| UAS-CG4733[i] | VDRC | v34894 |
| UAS-CG14395[i] | VDRC | v17517 |
| **Antibodies** | | |
| Anti-dorsal antibody (*Drosophila*) | DSHB | 7A4 |
| Anti-mouse Alexa Fluor 633 | Thermo Fisher Scientific | A-21126 |
| Anti-mouse Alexa Fluor 546 | Thermo Fisher Scientific | A-11004 |
| DAPI-containing SlowFade Gold Antifade Reagent | Molecular Probes | S36939 |
| Anti-β-catenin (human) | Dako, CA, USA | M3539 |

| Reagent/resource | Reference or source | Identifier or catalogue number |
|---|---|---|
| Anti-IKKα (human) | Genway, CA, USA | GWB-662250 |
| Anti-IKKβ (human) | Abcam, Cambridge, UK | ab32135 |
| Anti-IKKα$^{s176}$ (human) | Abcam, Cambridge, UK | ab138426 |
| **Oligonucleotides and other sequence-based reagents** | | |
| TRIzol® | Invitrogen™, Life Technologies | 15596018 |
| iScriptTM gDNA Clear cDNA Synthesis Kit | Bio-Rad Laboratories Ltd | 1725035 |
| iTaq™ Universal SYBR® Green Supermix kit | Bio-Rad Laboratories Ltd | 1725124 |
| Rp49_forward: CGCTTCAAGGGACAGTATCTG | Suzawa et al, 2019 | N/A |
| Rp49_reverse: AAACGCGGTTCTGCATGA | Suzawa et al, 2019 | N/A |
| Drosomycin_forward: CTCTTCGCTGTCCTGATGCT | Kleino et al, 2017 | N/A |
| Drosomycin_reverse: ACAGGTCTCGTTGTCCCAGA | Kleino et al, 2017 | N/A |
| **Chemicals, enzymes and other reagents** | | |
| Trametinib | Selleckchem or biorbyt | S2673 or ORB546250-BOR |
| QNZ (EVP4593) | Selleckchem | S4902 |
| iCRT3 | Selleckchem | S8647 |
| IWP-01 | Selleckchem | S8645 |
| XAV-939 | Selleckchem | S1180 |
| Capmatinib | Selleckchem | S2788 |
| JSH-23 | Selleckchem | S7351 |
| PNU-74654 | Selleckchem | S8429 |
| Propidium iodide | Selleckchem | S6874 |
| CellTiter-Fluor (TM) Cell Viability Assay | Promega | G6082 |
| UDP-Glo™ Glycosyltransferase Assay | Promega | V6961 |
| **Software** | | |
| RStudio | Posit | N/A |
| GPT-3.5 | Open AI | N/A |
| GraphPad Prism 10 | GraphPad Software | N/A |
| Microsoft Word | Microsoft | N/A |
| Microsoft PowerPoint | Microsoft | N/A |
| **Other** | | |
| T84 | ATCC | CCL-248™ |
| SW620 | ATCC | CCL-227 |

### *Drosophila* strains and genetics

Fly lines were cultured at room temperature or 25–29 °C on standard fly food or food-plus-compound. Fly food contained agar 10 g, soya flour 5 g, sucrose 15 g, glucose 33 g, maize meal 15 g, wheat germ 10 g, treacle molasses 30 g, yeast 35 g, nipagin 10 ml, propionic acid 5 ml in 1000 ml water. Transgenes used (Bloomington Drosophila Stock Centre number): *byn-gal4* (hindgut-specific line, V. Hartenstein), *UAS-Ras$^{G12V}$* (second chromosome, G. Halder), *tub-gal80$^{TS}$* (#7017), *w$^{1118}$* (#3605), *UAS-mCD8-GFP* (#5137), *UAS-dl$^i$* (#36650), *UAS-dif$^i$* (#30513), *UAS-cact$^i$* (#37484), UAS-Arm$^{S10}$ (#4782), *UAS-Toll1$^i$* (#35628), *UAS-Toll3$^i$* (#28526), *UAS-Toll4$^i$* (#28543), *UAS-Toll5$^i$* (#29533), *UAS-Toll6$^i$* (#56048), *UAS-Toll7$^i$* (#30488), *UAS-Toll8$^i$* (#28519), *UAS-Toll9$^i$* (#34853), *UAS-brm$^i$* (#35211), *UAS-shg$^i$* (#38207), *UAS-ago$^i$* (#34802), *UAS-rhoGAPp190$^i$* (#43987), *UAS-upf1$^i$* (#64519), *UAS-CG13344$^i$* (#41831), *UAS-p38a$^i$* (#35244), *UAS-ft$^i$* (#34970), *UAS-put$^i$* (#39025), *UAS-dnapol-eta$^i$* (#33410), *UAS-lrp1$^i$* (#44579), *UAS-tefu$^i$* (#44073), *UAS-nej$^i$* (#37489), *UAS-nos$^i$* (#33973), *UAS-pc$^i$* (#36070), *UAS-rad51c$^i$* (#67355), *dl[1]* (#3236), *UAS-blanks$^i$* (#33667), *UAS-cht4$^i$* (#65001), *UAS-mfs14$^i$* (#33999), *UAS-cht5$^i$* (#57512), *UAS-ref(2)p$^i$* (#36111), *UAS-CG32302$^i$* (#67239), *UAS-CG17104$^i$* (#42925), *UAS-mec2$^i$* (#61259), *UAS-CG15739$^i$* (#57216), *UAS-ag5r$^i$* (#67225), *UAS-arc1$^i$* (#25954), *UAS-CG2065$^i$* (#55283), *UAS-CG10182$^i$* (#61954), *UAS-CG18473$^i$* (#57524), *UAS-cdc23$^i$* (#61982), *UAS-rpt3r$^i$* (#58140), *UAS-ter94$^i$* (#35608), *UAS-prosalpha4$^i$* (#65161), *UAS-alh$^i$* (#39057), *UAS-punch$^i$* (#41998), *UAS-CG4502$^i$* (#35489), *UAS-CG12493$^i$* (#42791), *UAS-vis$^i$* (#35738), *UAS-rpt4$^i$* (#32874), *UAS-rpn3$^i$* (#34561), *UAS-cyp4p1$^i$* (#67349), *UAS-hsp23$^i$* (#82961), *UAS-mal-a6$^i$* (#60398), *UAS-fng$^i$* (#25947), *UAS-CG30427$^i$* (#58271), *UAS-prosbeta3$^i$* (#34868), *UAS-idh$^i$* (#41708), *UAS-CG8036$^i$* (#60371), *UAS-nd-pdsw$^i$* (#29592), *UAS-cda4$^i$* (#65909), *UAS-muc$^i$* (#44439), *UAS-CG4459$^i$* (#61228), *UAS-cox5a$^i$* (#58282), *UAS-CG32564$^i$* (#58342), *UAS-nd-b22$^i$* (#65011), *UAS-cox7a$^i$* (#57572), *UAS-rel$^i$* (#33661), *UAS-domeHMJ21208$^i$* (#53890), *UAS-domeHMS10293$^i$* (#34618), *UAS-bsk$^i$* (#36643), *UAS-CG9360$^i$* (Vienna Drosophila Resource Centre, #v13189), *UAS-CG1698$^i$* (#v101947), *UAS-CG32365$^i$* (#v104119), *UAS-CG4733$^i$* (#v34894), *UAS-CG14395$^i$* (#v17517).

### Chemicals

Drugs and compounds were used as follows: trametinib (Selleckchem or Biorbyt), QNZ (EVP4593), iCRT3, IWP-01, XAV-939, Capmatinib, JSH-23, PNU-74654 and propidium iodide (PI) purchased from Selleckchem. Drug and compound stocks were diluted in DMSO or water; drugs were then mixed into standard fly food with final DMSO concentration 0.1% to prevent toxicity.

### Statistical analysis

Eggs were collected for 24 h in drug-containing food at 18 °C to minimise transgene expression during embryogenesis to prevent embryonic effects or lethality. After 3 days, the tubes were transferred to the appropriate temperature (25–29 °C) to induce transgene expression; the number of surviving Drosophila adults was quantified after 2 weeks. *w$^{1118}$* served as a control in this study. Transgenic flies and non-transgenic flies' pupae (TP and no-TP (endogenous control)), respectively, were counted for each test tube, and the percentage of adult survival to control was calculated using the formula [(TP/no-TP) × 100]. Each point on the survival graph represents data from a test tube.

Statistical analysis was performed using Prism 10. Based on the examined statistical distribution and variance, we have chosen the appropriate statistical test, e.g. if the data were not normally distributed and multiple comparisons were required, we used a nonparametric statistical test (e.g. "Dunnett's multiple comparisons

test"). N.S $P(>0.12)$, $*P(0.033)$, $**P(0.002)$ and $***P(<0.001)$. All experiments were reproduced at least three times. All statistical data are summarised in Table EV1. All detailed genotypes are summarised in Table EV2.

## Imaging of the digestive tract of third instar larvae

Third instar larvae were dissected in 1x PBS and fixed with 4% paraformaldehyde for 30 min at room temperature, then washed $3 \times 15$ min in PBT (0.1% Triton X in 1x PBS). Samples were incubated in anti-dorsal primary antibody (#7A4, DSHB, 1:100); the secondary antibody used was anti-mouse Alexa Fluor 546 or 633 (Invitrogen, 1:250). Samples were mounted with DAPI-containing SlowFade Gold Antifade Reagent (#S36939, Molecular Probes). Fluorescence images were visualised on a Leica TSC SPE confocal microscope.

## RNA isolation and quantitative real-time PCR (*Drosophila*)

Total RNA from 30 hindguts was isolated using TRIzol® according to the manufacturer's protocol (cat.15596018, Invitrogen™, Life Technologies). mRNA was reverse transcribed using iScript™ gDNA Clear cDNA Synthesis Kit (cat# 1725035, Bio-Rad Laboratories Ltd).

For quantitative Real-Time PCR (qPCR), iTaq™ Universal SYBR® Green Supermix kit (cat. #1725124, Bio-Rad Laboratories Ltd.) was used according to the manufacturer's recommendation with cDNA (diluted 1:10–20) as a template. RT-qPCRs were performed with three biological replicates. Relative expression values were determined by the $2^{-\Delta\Delta Ct}$ method using *rp49* as an endogenous control. The RT-qPCR primers used are as follows: *rp49* (forward: CGCTTCAAGGGACAGTATCTG; reverse: AAACGCGGTTCTGCATGA), *drosomycin* (forward: CTCTTCGC TGTCCTGATGCT; reverse: ACAGGTCTCGTTGTCCCAGA).

## RNA isolation and RNA sequencing (*Drosophila*)

RNA sequencing was run and analysed by the CRUK Beatson Institute. The reference genome used was Drosophila melanogaster. BDGP6.46.110 (Ensembl genome). Reads were quality checked using FastQC version 0.11.8 and then trimmed with TrimGalore version 0.6.4 to remove adaptors and low-quality reads (Phred score <20). Then, aligned to the reference above using Hisat2 version 2.1.0, and gene-level counts were determined using FeatureCounts version 1.6.4. The differential expression analyses were done in R using DESeq2 version 1.22.2, which uses a Wald test to assess significance between groups. Graphs were drawn by using ggplot2 package of R, downregulated genes marked by blue (adjusted $p$ values <0.05 and log2(fold change <−0.3); upregulated genes marked by red (adjusted $p$ values <0.05 and log2(fold change >0.3).

## Endogenously released UDP assay (*Drosophila*)

Eggs were collected for 24 h in drug-containing food at 18 °C to minimise transgene expression during embryogenesis to prevent embryonic effects or lethality. After 3 days, the tubes were transferred to the appropriate temperature to induce transgene expression. After 4 days, third instar larvae were dissected in 1x

PBS, and the hindgut was assayed in 1x PBS. Cell number (CN) was measured by CellTiter-Fluor™ Cell Viability Assay kit (Promega). Total endogenous UDP release (TEUDP) in the hindgut was measured by UDP-Glo™ Glycosyltransferase Assay kit (Promega). The released UDP level in each cell was calculated using the formula [TEUDP/CN].

## Immunohistochemistry for detection of β-catenin, IKKβ, IKKα and phospho-IKKα^s176

Samples from a retrospective cohort of 787 stage 2–3 colorectal cancer patients were stained via immunohistochemistry (IHC) for β-catenin, IKKβ, IKKα and phospho-IKKα serine 176 (IKKα^s176). Staining was performed on a previously constructed tissue microarray (TMA), which consisted of CRC tissue from patients undergoing surgery with curative intent within Greater Glasgow and Clyde hospitals between 1997 and 2013. Data were stored within the Glasgow Safehaven (GSH21ON009), and ethical approval was in place for the study (MREC/01/0/36).

IHC was performed as previously described (Al-Badran et al, 2021). Briefly, TMA sections were dewaxed and then rehydrated through a series of alcohols. Antigen retrieval was performed using citrate buffer (pH 6) for β-catenin, IKKβ and IKKα, and Tris EDTA (pH 9) for IKKα^s176. Endogenous peroxidases were blocked in 3% hydrogen peroxide. Tissue was blocked using 10% casein (SP-5020, Vector Laboratories, CA, USA) for β-catenin and IKKα^s176, and 5% horse serum (S-2000, Vector Laboratories, CA, USA) for IKKβ and IKKα, incubating for 1 h at room temperature. Sections were incubated in primary antibody β-catenin ((M3539, Dako, CA, USA, 1:600), IKKα (GWB-662250, Genway, CA, USA, 1:4000), IKKβ (ab32135, Abcam, Cambridge, UK, 1:200) and IKKα^s176(ab138426, Abcam, Cambridge, UK, 1:150) overnight at 4 °C. Sections were washed in tris-buffered saline (TBS), incubated in Impress secondary antibody (MP-7500, Vector Laboratories, CA, USA) for 2 h at room temperature. Sections were washed in TBS and incubated for 5 min in 3,3′-diaminobenzidine (DAB) (SK-4105, Vector Laboratories, CA, USA). Slides were rinsed in water, counterstained and dehydrated before mounting with Pertex (00801-EX, Histolab products, Askim, Sweden). Stained sections were imaged using a Hamamatsu NanoZoomer (Hamamatsu Photonics, Shizuoka, Japan) onto an NZ Connect viewing platform (Hamamatsu Photonics, Shizuoka, Japan).

## Staining quantification of β-catenin, IKKβ, IKKα and phospho-IKKα^s176 (human CRC)

Staining intensity was assessed semi-quantitatively by weighted histoscore using QuPath® software in the tumour cell cytoplasm for β-catenin, IKKβ, IKKα and IKKα^s176 (Bankhead et al, 2017). Continuous scores ranging from 0 to 300 for β-catenin were dichotomised into high and low expression groups using the Survminer package in RStudio (version 1.4, RStudio, Boston, MA, USA).

## Mutational profiling and analysis (human CRC samples)

CRC tissue from the patient cohort was profiled for the presence of KRAS mutation by BioClavis (BioClavis Ltd, Glasgow, UK). Patients were grouped into three categories based on KRAS status and β-catenin expression. Group 1 patients were wild type for

KRAS and low for β-catenin, Group 2 patients were either KRAS mutant or high for β-catenin and Group 3 patients were both KRAS mutant and high for β-catenin. These groups were then assessed for association with IKKβ, IKKα and IKKα$^{s176}$ expression using T-tests in GraphPad Prism (GraphPad Software, La Jolla, CA, USA).

## Endogenously released UDP assay and cell viability assay (colon cancer cells)

Seed 3000 cells per well (T84, ATCC, CCL-248™ or SW620, ATCC, CCL-227) in a 96-well plate for each cell line. After 24 h, treat the cells with the respective drugs or 0.1% DMSO as a control. Incubate for 48, 96 and 144 h under standard conditions. At each time point, assess cell viability (CV) by adding CellTiter-Fluor Reagent according to the manufacturer's instructions. To evaluate total endogenous UDP release (TEUDP), used the UDP-Glo™ Glycosyl-transferase Assay Kit (Promega). The released UDP levels for each condition were calculated using the formula [TEUDP/CV].

## Use of a large language model

Some sentences were revised with the aid of GPT-3.5, strictly to improve clarity. The authors take full responsibility for the accuracy of all prose in the manuscript.

## Data availability

No primary datasets have been generated or deposited.

The source data of this paper are collected in the following database record: biostudies:S-SCDT-10_1038-S44319-025-00588-1.

## Peer review information

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

## Acknowledgements

We thank the Cagan and Edwards Laboratory members for their important discussions. We also thank the Bloomington Drosophila Stock Centre for Drosophila stocks, the Glasgow Tissue Research Facility, the NHS Glasgow Biorepository and the Academic Unit of Surgery at the Glasgow Royal Infirmary. We also thank the patients who donated tissue samples that were utilised in this study. This work was generously supported by grants from the NIH (R01CA258736), a Royal Society Wolfson Fellowship, Chief Scientific Office (EPD/22/13) (TCS/22/02), CRUK (CTRQQR-2021\100006) and Beatson Cancer Charity (24-25-045 BCC).

## Author contributions

**Bojie Cong**: Conceptualisation; Data curation; Formal analysis; Investigation; Methodology; Writing—original draft; Writing—review and editing. **Evangelia Stamou**: Investigation. **Kathryn Pennel**: Data curation; Formal analysis; Investigation. **Teena Thakur**: Formal analysis; Investigation. **Molly McKenzie**: Data curation; Formal analysis; Investigation. **Amna Matly**: Data curation; Formal analysis; Investigation. **Kathryn Gilroy**: Data curation; Formal analysis; Investigation. **Harshit Shah**: Data curation; Formal analysis; Investigation. **Sindhura Gopinath**: Resources. **Joanne Edwards**: Conceptualisation; Data curation; Supervision; Project administration. **Ross Cagan**: Conceptualisation; Supervision; Funding acquisition; Writing—original draft; Project administration; Writing—review and editing.

Source data underlying figure panels in this paper may have individual authorship assigned. Where available, figure panel/source data authorship is listed in the following database record: biostudies:S-SCDT-10_1038-S44319-025-00588-1.

## Disclosure and competing interests statement

The authors declare no competing interests.

# Expanded View Figures

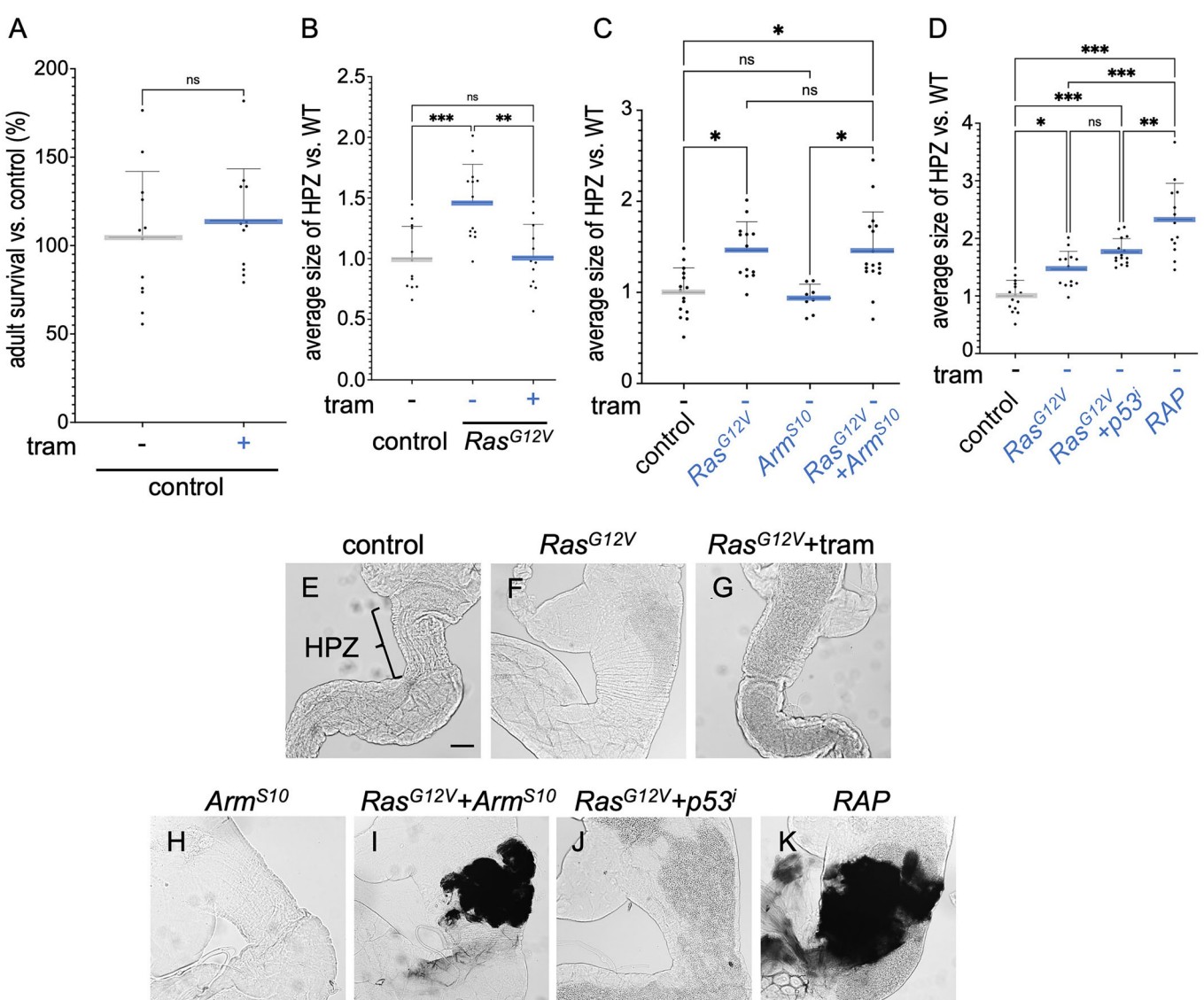

**Figure EV1. Overgrowth of HPZ was driven by combined Ras, Wg and p53 alterations.**

(A) Percent survival of control flies to adulthood relative to control flies was quantified in the presence or absence of trametinib (1 μM). Control (DMSO, n = 12; tram, n = 12). (B–D) The average of the hindgut proliferation zone (HPZ) size was measured by Fiji ImageJ and quantified as relative size to the control hindgut. Control (DMSO, n = 12), Ras$^{G12V}$ (DMSO, n = 13; tram, n = 12) (B); Control (DMSO, n = 15), Ras$^{G12V}$ (DMSO, n = 13), Arm$^{S10}$ (DMSO, n = 8), Ras$^{G12V}$;Arm$^{S10}$ (DMSO, n = 17) (C); Control (DMSO, n = 15), Ras$^{G12V}$ (DMSO, n = 13), RAP (DMSO, n = 13), Ras$^{G12V}$;p53$^i$ (DMSO, n = 15) (D). (E–K) Images of the digestive tract of third instar larvae in the presence or absence of trametinib (1 μM). Control (E), Ras$^{G12V}$ (F, G), Arm$^{S10}$ (H), Ras$^{G12V}$;Arm$^{S10}$ (I), Ras$^{G12V}$;p53$^i$ (J) and RAP (K). Scale bar 200 μm. (A–K) The experiment was conducted at 27 °C. The statistical tests used to calculate the P value are as follows: (A) Mann–Whitney test; (B–D) one-way ANOVA; NS P(>0.12), *P(0.033), **P(0.002) and ***P(<0.001). All statistical data are summarised in Table EV1. The error bar is a standard deviation (SD), with each point representing biological replicates and numbers (n), including three technical replicates.

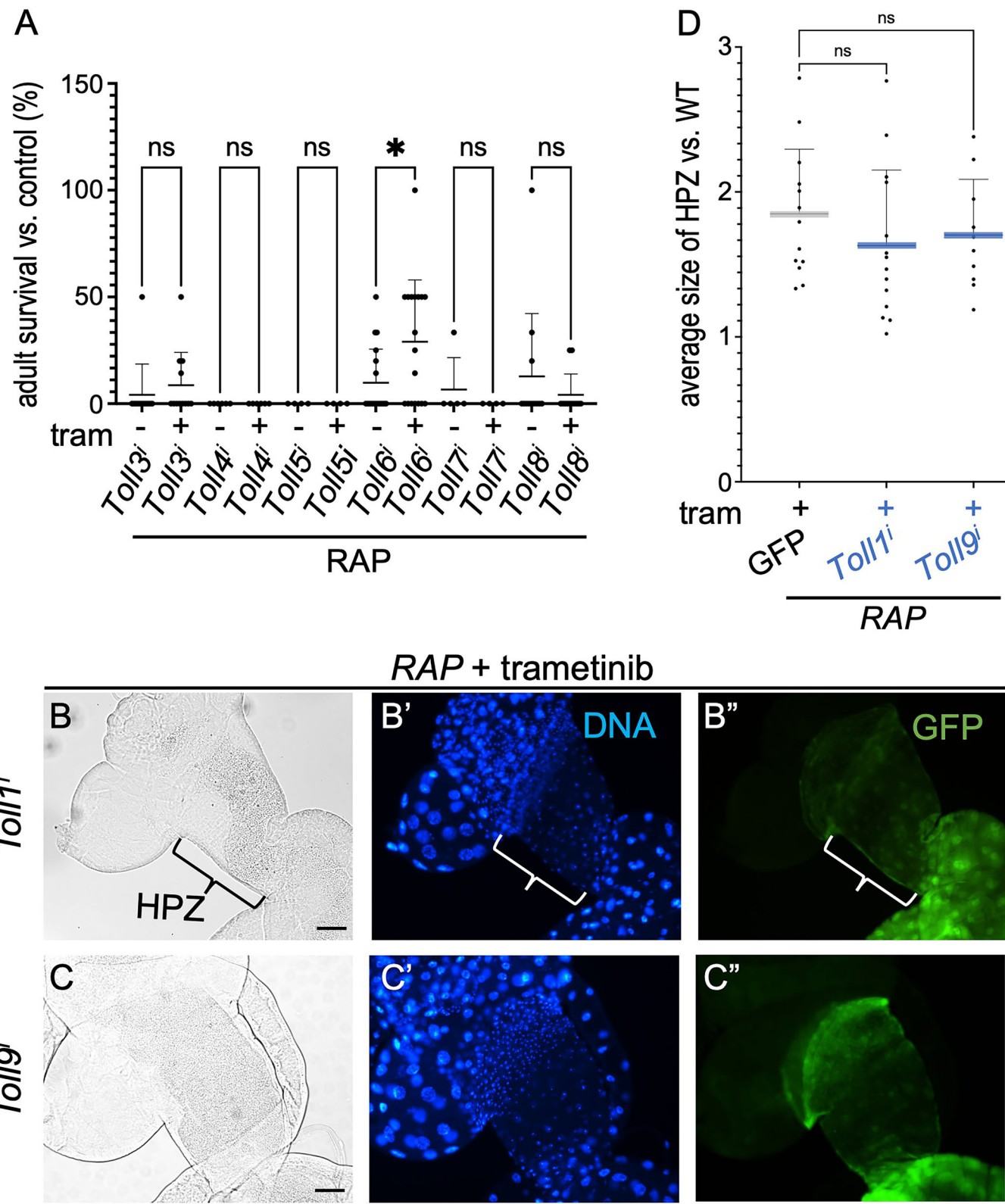

◀ **Figure EV2.  Administration of NF-κB inhibitors did not suppress drug resistance in *RAP* tumours.**

(A) Percent survival of transgenic flies to adulthood relative to control flies was quantified in the presence or absence of trametinib (1 μM), *RAP;Toll3[i]* (DMSO, *n* = 12; tram, *n* = 12), *RAP;Toll4[i]* (DMSO, *n* = 6; tram, *n* = 6), *RAP;Toll5[i]* (DMSO, *n* = 4; tram, *n* = 4), *RAP;Toll6[i]* (DMSO, *n* = 18; tram, *n* = 18), *RAP;Toll7[i]* (DMSO, *n* = 5; tram, *n* = 4), *RAP;Toll8[i]* (DMSO, *n* = 12; tram, *n* = 12). (B, C). Images of the digestive tract of third instar larvae in the presence of trametinib (1 μM), Scale bar 200 μm. (D) Average hindgut proliferation zone (HPZ) size was measured by Fiji ImageJ and quantified as relative size to the control hindgut. *RAP;GFP* (tram, *n* = 13), *RAP;Toll1[i]* (tram, *n* = 14), *RAP;Toll9[i]* (tram, *n* = 10). (A–D) The experiment was conducted at 29 °C. The statistical tests used to calculate the *P* value are as follows: (A, B) one-way ANOVA; NS *P*(>0.12), *\*P*(0.033), *\*\*P*(0.002) and *\*\*\*P*(<0.001). All statistical data are summarised in Table EV1. The error bar is a standard deviation (SD), with each point representing biological replicates and numbers (*n*), including three technical replicates.

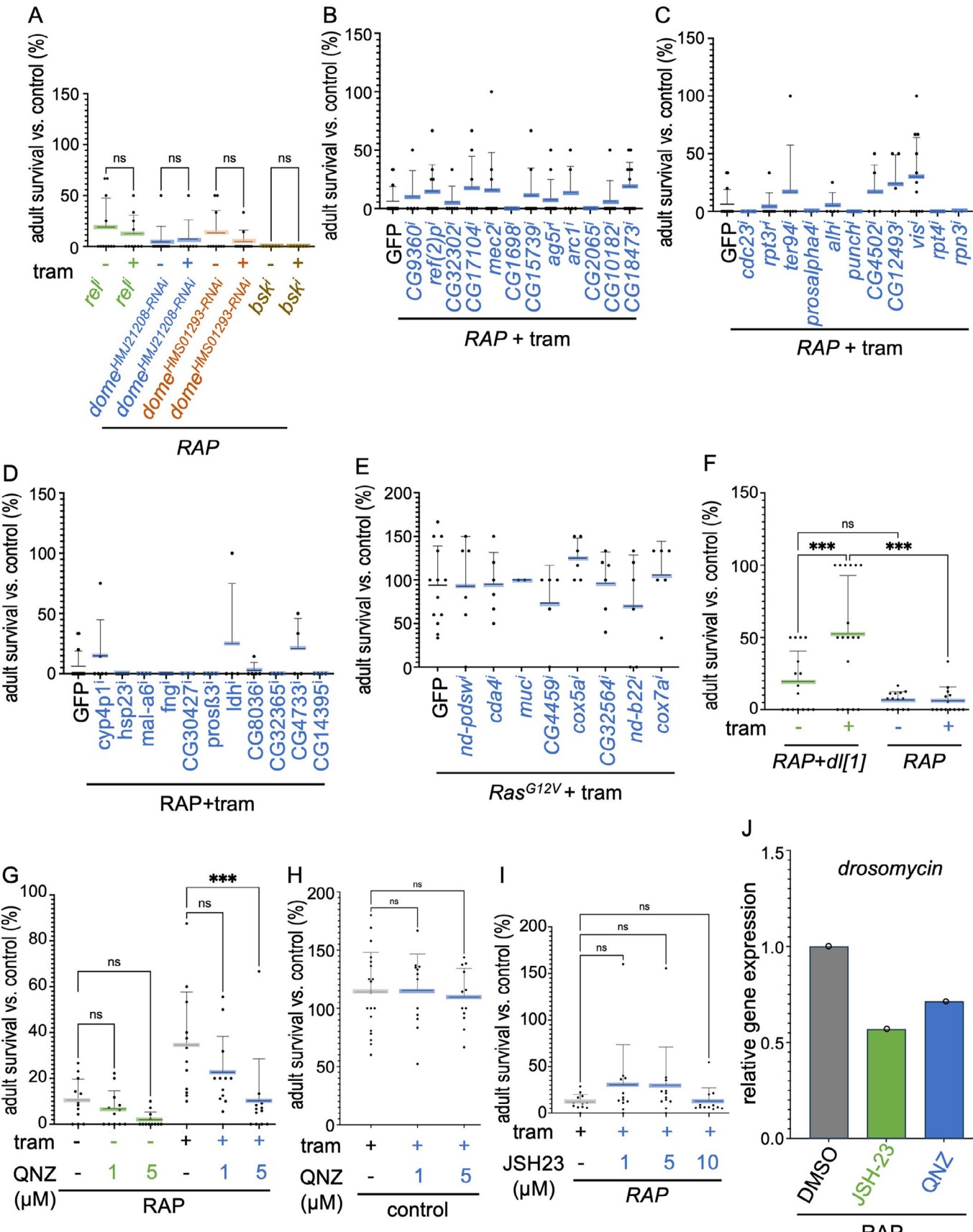

◀ **Figure EV3. Screening for factors that influence drug response.**

(A–E) Percent survival of transgenic flies to adulthood relative to control flies was quantified in the presence or absence of trametinib (1 μM). All flies driven by *byn-GAL4*: (A) *RAP;rel*[i] (DMSO, *n* = 11; tram, *n* = 10), *RAP;dome*[i] (HMJ21208) (DMSO, *n* = 11; tram, *n* = 7), *RAP;dome*[i] (HMS01293) (DMSO, *n* = 10; tram, *n* = 10), *RAP;bsk*[i] (DMSO, *n* = 6; tram, *n* = 6); (B) *RAP;GFP* (tram, *n* = 14), *RAP;CG9360*[i] (tram, *n* = 5), *RAP;ref(2)p*[i] (tram, *n* = 10), *RAP;CG32302*[i] (tram, *n* = 6), *RAP;CG17104*[i] (tram, *n* = 8), *RAP;mec2*[i] (tram, *n* = 10), *RAP;CG1698*[i] (tram, *n* = 5), *RAP;CG15739*[i] (tram, *n* = 9), *RAP;ag5r*[i] (tram, *n* = 11), *RAP;arc*[i] (tram, *n* = 6), *RAP;CG2065*[i] (tram, *n* = 6), *RAP;CG10182*[i] (tram, *n* = 8), *RAP;18473*[i] (tram, *n* = 12); (C) *RAP;GFP* (tram, *n* = 14), *RAP;cdc23*[i] (tram, *n* = 3), *RAP;rpt3r*[i] (tram, *n* = 8), *RAP;ter94*[i] (tram, *n* = 6), *RAP;prosalpha4*[i] (tram, *n* = 4), *RAP;alh*[i] (tram, *n* = 5), *RAP;punch*[i] (tram, *n* = 6), *RAP;CG4502*[i] (tram, *n* = 5), *RAP;CG12493*[i] (tram, *n* = 5), *RAP;vis*[i] (tram, *n* = 12), *RAP;rpt4*[i] (tram, *n* = 5), *RAP;rpn3*[i] (tram, *n* = 2); (D) *RAP;GFP* (tram, *n* = 14), *RAP;cyp4p1*[i] (tram, *n* = 6), *RAP;hsp23*[i] (tram, *n* = 6), *RAP;mal-a6*[i] (tram, *n* = 4), *RAP;fng*[i] (tram, *n* = 6), *RAP;CG30427*[i] (tram, *n* = 3), *RAP;prosbeta3*[i] (tram, *n* = 4), *RAP;ldh*[i] (tram, *n* = 4), *RAP;CG8036*[i] (tram, *n* = 5), *RAP;CG32365*[i] (tram, *n* = 4), *RAP;CG4733*[i] (tram, *n* = 4), *RAP;CG14395*[i] (tram, *n* = 4); (E) *Ras*[G12V];*GFP* (tram, *n* = 13), *Ras*[G12V];*nd-pdsw*[i] (tram, *n* = 6), *Ras*[G12V];*cda4*[i] (tram, *n* = 6), *Ras*[G12V];*muc*[i] (tram, *n* = 2), *Ras*[G12V];*CG4459*[i] (tram, *n* = 5), *Ras*[G12V];*cox5a*[i] (tram, *n* = 6), *Ras*[G12V];*CG32564*[i] (tram, *n* = 6), *Ras*[G12V];*nd-b22*[i] (tram, *n* = 6), *Ras*[G12V];*cox7a*[i] (tram, *n* = 6). (F–I) Percent survival of transgenic flies to adulthood relative to control flies was quantified in the presence or absence of trametinib (1 μM), QNZ (EVP4593) or JSH-23. (F) *RAP;dl[1]* (DMSO, *n* = 16; tram, *n* = 18) and *RAP* (DMSO, *n* = 14; tram, *n* = 14); (G–I) *RAP* (DMSO, *n* = 12; tram, *n* = 12; 1 μM QNZ, *n* = 12; 5 μM QNZ, *n* = 12; tram + 1 μM QNZ, *n* = 12; tram + 5 μM QNZ, *n* = 12) (G); control (tram, *n* = 20; tram + 1 μM QNZ, *n* = 12; tram + 5 μM QNZ, *n* = 12) (H); *RAP* (tram, *n* = 12; tram + 1 μM JSH-23, *n* = 12; tram + 5 μM JSH-23, *n* = 12; tram + 10 μM JSH-23, *n* = 12) (I). (J) Expression levels of *drosomycin* were quantified for each condition by quantitative RT-PCR, QNZ (5 μM), JSH-23 (5 μM). (A–D, F) The experiments were conducted at 29 °C. (E, G–I) The experiments were conducted at 29 °C. The statistical tests used to calculate the *P* value are as follows: (A, F–I) one-way ANOVA; NS *P*(>0.12), *P*(0.033) and **P*(<0.001). All statistical data are summarised in Table EV1. The error bar is a standard deviation (SD), with each point representing biological replicates and numbers (*n*), including three technical replicates.

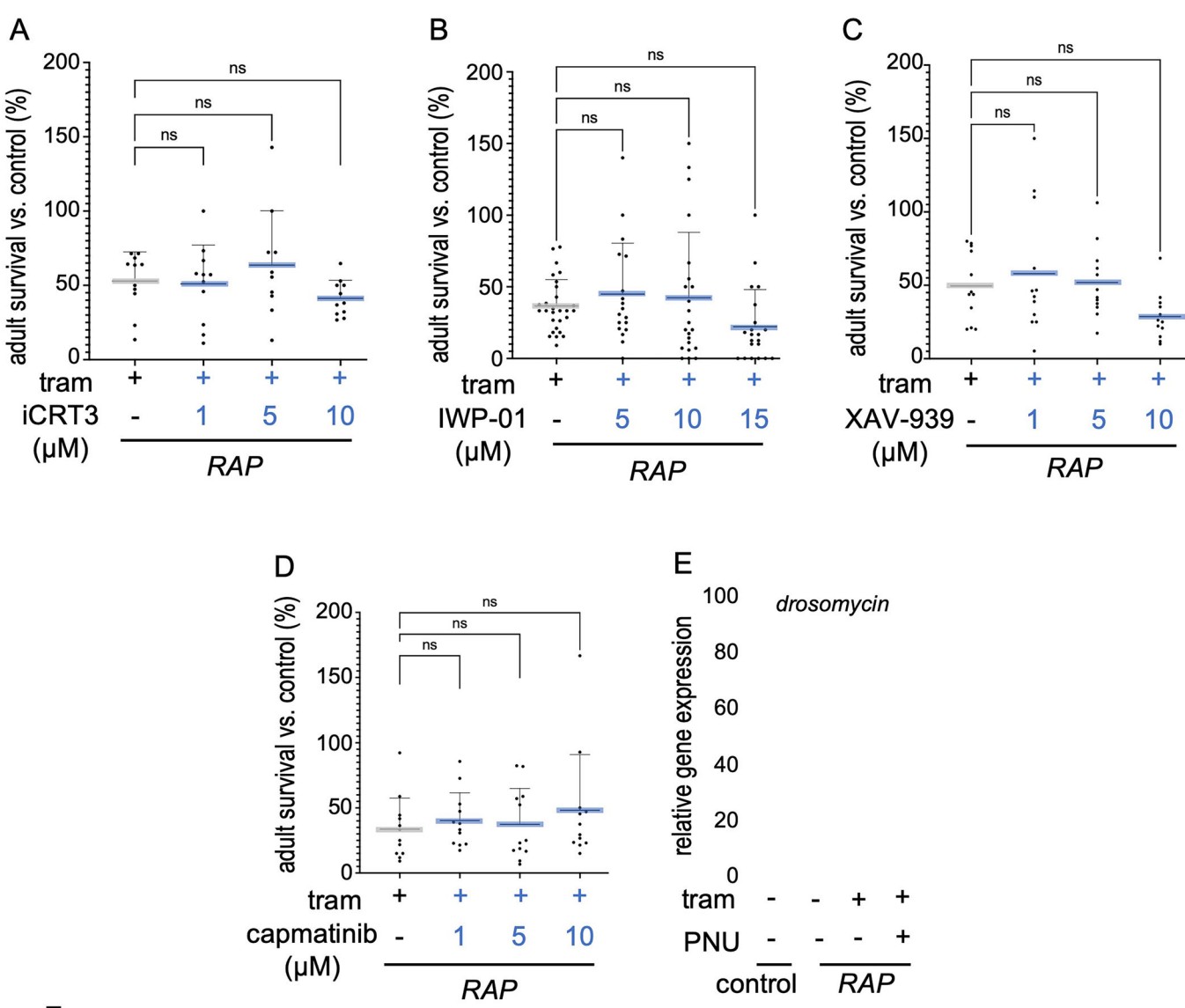

| Line | Gain of function genes | Loss of function genes |
|------|------------------------|------------------------|
| CPCT006 | *Ras^G12V* | *apc; p53; ago; shg; put; p38a; ft; brm* |
| CPCT018 | *Ras^G12V* | *apc; p53; pten; smox; tefu; rhoGAPp190; nejire; upf1* |
| CPCT029 | *Ras^G12V; chico* | *apc; p53; Med; smox; trr; fur2; CG4238; PI4KIIIalpha, mus81* |
| CPCT045 | *Ras^G12V; pvr* | *apc; p53; smox; nos; CG13344; pc; rad51C; DNApol-eta; lrp1* |
| CPCT050 | *Ras^G12V* | *apc; p53; debcl; scat; wdb; ird1* |
| RAPp1 | *Ras^G12V* | *apc; p53; ago; wts; CG7742; Atg2* |
| RAPp2 | *Ras^G12V* | *apc; p53; vrp1; ry; khc-73* |

Figure EV4.   A screening for trametinib and WNT inhibitors drug combination in *RAP* hindgut tumours.

(A–D) Percent survival of transgenic *RAP* flies to adulthood relative to control flies was quantified in the presence or absence of trametinib (1 μM), iCRT3, IWP-01, XAV-939 or Capmatinib. *RAP* (tram, $n = 11$; tram + 1 μM iCRT3, $n = 11$; tram + 5 μM iCRT3, $n = 10$; tram + 10 μM iCRT3, $n = 11$) (A); *RAP* (tram, $n = 28$; tram + 5 μM IWP-01, $n = 18$; tram + 10 μM IWP-01, $n = 22$; tram + 15 μM IWP-01, $n = 21$) (B); *RAP* (tram, $n = 12$; tram + 1 μM XAV-939, $n = 12$; tram + 5 μM XAV-939, $n = 12$; tram + 10 μM XAV-939, $n = 12$) (C); *RAP* (tram, $n = 12$; tram + 1 μM Capmatinib, $n = 12$; tram + 5 μM Capmatinib, $n = 12$; tram + 10 μM Capmatinib, $n = 12$) (D). (E) The expression levels of *drosomycin* among each genotype in the presence or absence of trametinib (1 μM) or PNU-74654 (1 μM) were detected by quantitative RT-PCR, $n = 3$. (E) *control* and *RAP*. (A–E) The experiment was conducted at 27 °C. The statistical tests used to calculate the P value are as follows: (A–D) one-way ANOVA; NS $P(>0.12)$, $*P(0.033)$, $**P(0.002)$ and $***P(<0.001)$. All statistical data are summarised in Table EV1. The error bar is a standard deviation (SD), with each point representing biological replicates and numbers (*n*), including three technical replicates. (F) A summary of mutations in patient-specific CRC models. These patient-specific fly avatars come from a fly-to-bedside study (CPCTs). and TCGA (*RAPp1, RAPp2*).

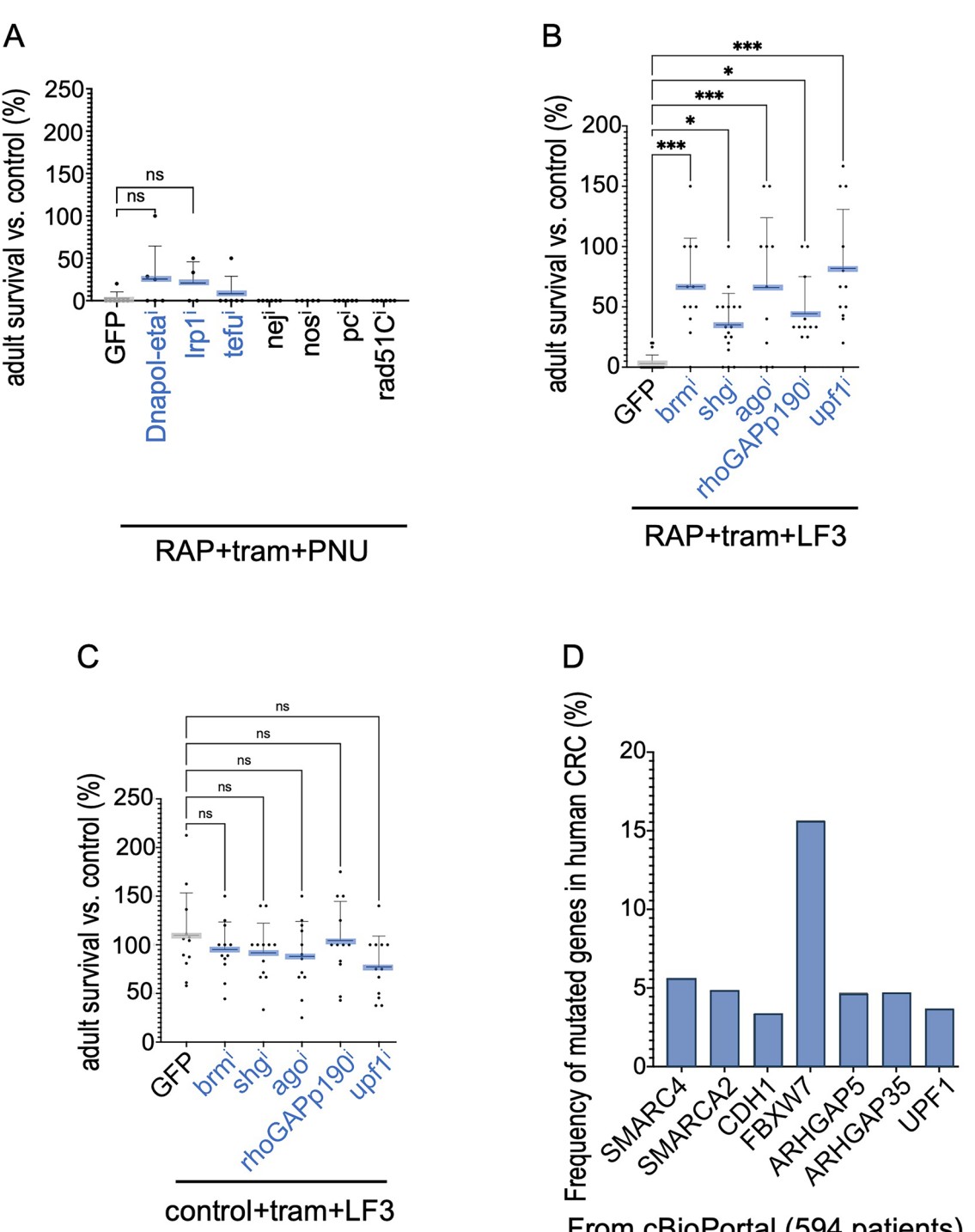

**Figure EV5. Regulators for the combination of trametinib and LF3 in CRC tumours.**

(A–C) Percent survival of transgenic flies to adulthood relative to control flies was quantified in the presence or absence of trametinib (1 μM), PNU-74654 (1 μM) or LF3 (10 μM). *RAP;GFP* (tram + PNU, n = 7), *RAP;DNApol-eta^i^* (tram+PNU, n = 6); *RAP;lrp1^i^* (tram + PNU, n = 4) (A). *RAP;GFP* (tram + LF3, n = 19), *RAP;brm^i^* (tram + LF3, n = 12), *RAP;ago^i^* (tram + LF3, n = 11), *RAP;shg^i^* (tram + LF3, n = 17), *RAP;upf1^i^* (tram + LF3, n = 12), *RAP;rhoGAPp190^i^* (tram + LF3, n = 12) (B); *control* (tram + LF3, n = 12), *brm^i^* (tram + LF3, n = 12), *ago^i^* (tram + LF3, n = 12), *shg^i^* (tram + LF3, n = 12), *upf1^i^* (tram + LF3, n = 12), *rhoGAPp190^i^* (tram + LF3, n = 12) (C). (D) The graph showed the frequency of mutated genes in human CRC. These data from cBioPortal include 594 patients. (A–C) The experiment was conducted at 29 °C. The statistical tests used to calculate the P value are as follows: (A–C) one-way ANOVA; NS P(>0.12), *P(0.033), **P(0.002) and ***P(<0.001). All statistical data are summarised in Table EV1. The error bar is a standard deviation (SD), with each point representing biological replicates and numbers (n), including three technical replicates.

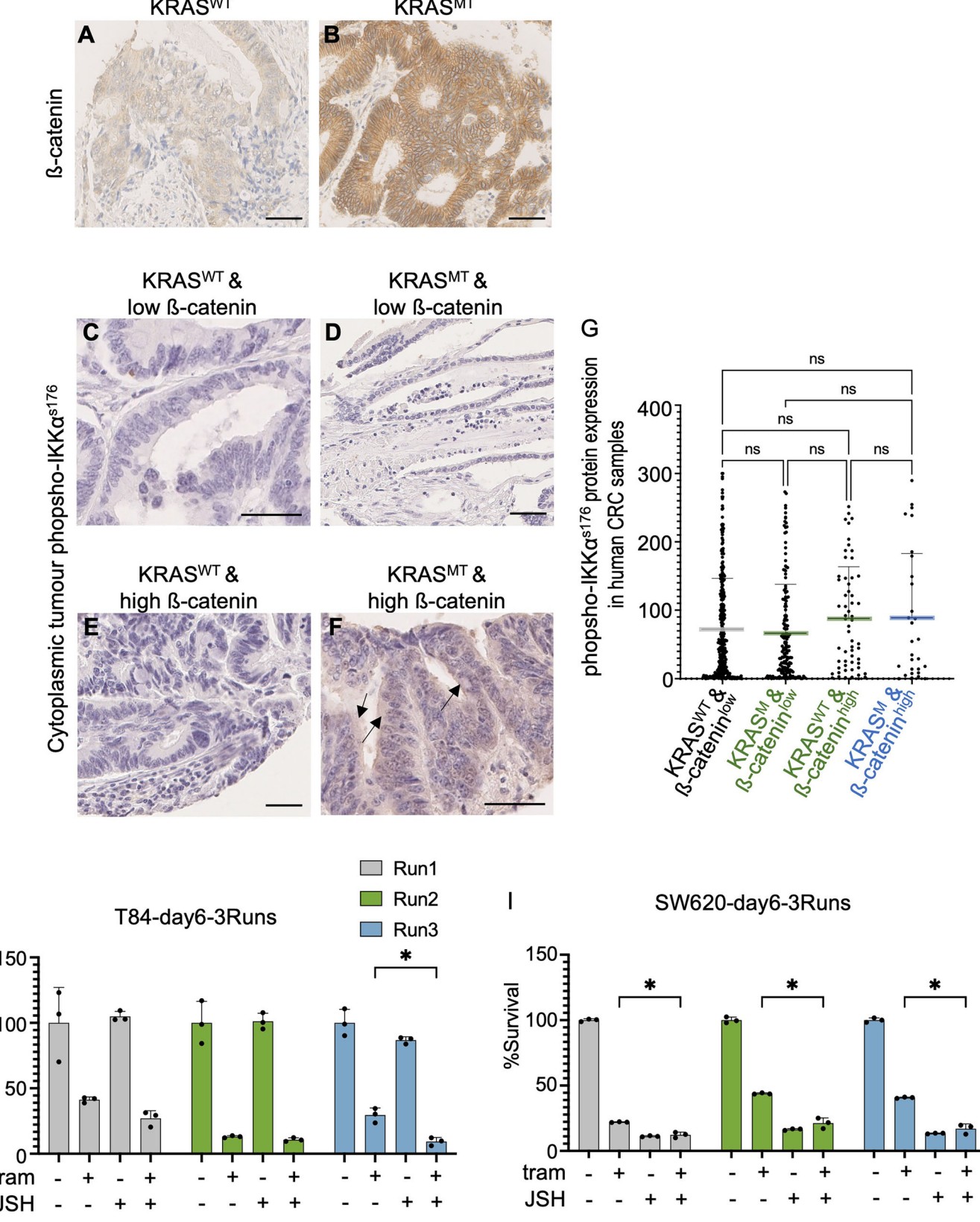

**Figure EV6. IHC staining of phospho-IKKα$^{s176}$ protein in human CRC.**

(A–F) IHC staining of β-catenin or phospho-IKKα$^{s176}$ in each genotype CRC sample. (A) *KRAS$^{WT}$* (wild-type); (B) *KRAS$^{MT}$* (mutation in position G12/G13); (C) *KRAS$^{WT}$* plus low β-catenin; (D) *KRAS$^{MT}$* plus low β-catenin; (E) *KRAS$^{WT}$* plus high β-catenin; (F) *KRAS$^{MT}$* plus high β-catenin expression in CRC samples. Scale bar 50 µm. (G) The graph shows the mean of expression of phospho-IKKα$^{s176}$ in each different mutated human CRC, determined by IHC intensity values. Patients were grouped into four categories based on KRAS status and β-catenin expression. Both *KRAS$^{WT}$* and β-catenin low ($n = 316$); *KRAS$^{M}$* and β-catenin low ($n = 164$); *KRAS$^{WT}$* and β-catenin high ($n = 59$); Both KRAS$^{M}$ and β-catenin high ($n = 29$). Each dot displays an individual sample. (H, I) Cell viability assay for T84 or SW620 colon cancer cells, showing the results of three experimental replicates, treatment with 0.1% DMSO, 50 nM trametinib or 10 µM JSH-23. The statistical tests used to calculate the *P* value are as follows: (G) one-way ANOVA; (H, I) 2-way ANOVA; NS $P (>0.12)$, $*P(0.033)$, $**P(0.002)$ and $***P(<0.001)$. All statistical data are summarised in Table EV1. The error bar is a standard deviation (SD), with each point representing biological replicates.

