## [Peer Review File · EMBO Reports]

WNT Signalling Promotes NF- κ B Activation and Drug Resistance in KRAS-Mutant Colorectal Cancer

Bojie Cong, Evangelia Stamou, Kathryn Pennel, Teena Thakur, Molly McKenzie, Amna Matley, Kathryn Gilroy, Harshit Shah, Sindhura Gopinath, Joanne Edwards, and Ross Cagan

Corresponding author(s): Ross Cagan (ross.cagan@glasgow.ac.uk)

Review Timeline:

Transfer Date:	18th Jul 25
Editorial Decision:	22nd Jul 25
Revision Received:	26th Aug 25
Accepted:	22nd Sep 25

Editor: Achim Breiling / Martina Rembold

Transaction Report: A revised version of this manuscript was transferred to EMBO reports following peer review at the EMBO Journal.

Referee #1:

The authors addressed all my concerns in a satisfactory way, and the MS is now ready for publication.

Referee #2:

Summary:

Cong et. al identify the NF- κ B pathway as a potential mediator of trametinib resistance in CRC tumors that express mutant RAS and activated Wnt. The authors show that inhibition of NF- κ B signaling correlates with a decrease in the levels of key glucuronidation and pentose pathway enzymes, decreased trametinib glucuronidation levels, and an increase in survival in their trametinib-resistant CRC Drosophila model. The authors describe the glucuronidation pathway as a detoxification pathway that can directly inactivate many cancer drugs, including trametinib. Thus, this study points to a very interesting role for the NF- κ B pathway in its potential promotion of the glucuronidation of trametinib.

Response:

The results showing that NF- κ B signaling affects key glucuronidation and pentose pathway enzymes, leading to decreased trametinib glucuronidation levels, is clear. However, the extent to which NF- κ B signaling plays a role in trametinib efficacy and survival is still unclear in this model.

Major Comments 1&3: There is still a major discrepancy in the data between the pharmacological and genetic targeting of NF- κ B, as these two methods lead to opposing results in trametinib-induced RAP lethality in vivo. The authors show that, "subtle reduction of dl (NF- κ B family member) was sufficient to strongly enhance the ability of trametinib to reduce RAP lethality, supporting canonical NF- κ B pathway components as adjunct therapies." Thus, genetic inhibition of NF- κ B leads to reduced RAP lethality under trametinib (S3F). The authors then employ clinical inhibitors of NF- κ B in combination with trametinib, but their data shows that this method of NF- κ B inhibition either increased RAP lethality, or had no effect in vivo. Thus, pharmacological inhibition of NF- κ B leads to increased RAP lethality under trametinib (S3G). These data are confounding. The authors kindly answered our previous question of whether these pharmacological inhibitors are functioning as expected, and they are, as these inhibitors reduced NF- κ B pathway activity as assessed with drosomycin expression. Therefore, these pharmacological inhibitors fail to promote drosophila survival, whereas genetic targeting of the NF- κ B pathway can promote drosophila survival, in the context of trametinib treatment of RAP tumors. The authors must acknowledge this discrepancy, because it challenges the overall conclusion

of the paper. In addition, the "inhibition of canonical NF- κ B activity by knockdown of dl or dif did not significantly suppress RAP tumour overgrowth in the presence of trametinib (compare Figure 142 1I-J to 1H, quantified in 1K)." Altogether, this data leads one to believe that the genetic models being used are affecting survival in ways that are independent of tumor biology, and that is why pharmacological targeting of this pathway does not produce the same effects on survival, and why tumor burden does not correlate with survival. The authors did not look into metastasis, toxicity, or specific forms of lethality that can explain these variations in survival as we suggested, but simply claim that whole body viability can be affected by NF- κ B signaling. Thus, these confounding results are not explained by the text or data.

Major Comment 4: We greatly appreciate the authors' investigation of these claims in human cell lines. These results show that JSH, the NF- κ B inhibitor that showed no effect on trametinib activity *in vivo*, can modestly enhance trametinib efficacy in one colon cancer cell line, but has a trametinib-independent function in another colon cancer cell line. Specifically, in T84 cells, JSH has no effect on its own, and the authors show a modest yet inconsistent enhancement cancer cell death when used in combination with trametinib. These results generally support the conclusion of the paper. In contrast, JSH employment in SW620 cells led to greater cancer cell death than trametinib alone, and the combination of trametinib and JSH was not greater than JSH treatment alone. Thus, JSH is functioning to reduce cancer cell survival independently of trametinib in SW620 cells. A dose-response curve of JSH would reveal whether the drug concentration is responsible for these variations, since differences are seen at baseline as a single agent, between the various cell lines. Inclusion of a third cell line would help decipher which outcome is more reproducible / relevant in CRC. Regardless, these results suggest context-dependent activity of JSH in CRC and in relation to trametinib, perhaps explaining its lack of efficacy *in vivo*.

Major Comment 2: The authors are fair in their assessment of the temporal implications of our question. The authors also provide a sufficient method of testing glucuronidation enhancement through UDP glucose administration within another study.

Minor Comment 1: The authors added a sentence broadly describing the pathways that were investigated in their initial screen. They did not mention any specific targets, or show any additional data. Supplying this information should have been easy, as they already have the data available.

Minor Comments 2&3: The authors provided sufficient explanations for their calculations of the percentage of adult survival. The authors added more detailed explanations of statistics within figure legends.

Overall, the authors answered some of our major questions pertaining to the manuscript. The authors must acknowledge the discrepancies between the survival effects of genetic

and pharmacological targeting of NF- κ B in the context of trametinib resistance. Proper discourse on this matter may be sufficient to include these confounding results. Outside of this, the authors addressed most of our questions.

Referee #3:

The manuscript has shown improvement and remains a strong study; however, several key concerns remain inadequately addressed, and it would be beneficial for the authors to clarify these points. Additionally, incorporating statistical analyses across key experiments to support or challenge their interpretations could significantly fortify their conclusions. The critical issue is the genetic and pharmacological impact of dif-RNAi or dl-RNAi. While it is understandable that determining the effects of host lethality without influencing tumorigenesis may be challenging at this stage, the implications of dif/dl upregulation and the role of TNF α discussed in relation to pharmacological inhibition remain vague. Further exploration of these aspects could offer valuable insights into potential new avenues for investigating resistance to trametinib.

Specifically, the issue of survival remains. In Figs 1 and 2 (e.g., 1C, 2A compared to 1A), it is evident that both dif-RNAi and dl-RNAi in the RAP tumours resulted in decreased survival compared to RAP alone (or RAP with tram) -it would be beneficial to compare the statistical significance of these groups. The impact of silencing Toll1 or Toll9 on host survival is less severe compared to the effects of silencing dif or dl. However, the data still pose similar interpretational challenges.

This discrepancy complicates the interpretation of the results, particularly since the tram+ RNAi did not achieve a survival level that allows for unambiguous interpretation of the interactions. As the degree of rescue conferred by tram treatment with both dl-RNAi or dif-RNAi does not (seem) to differ significantly from that of RAP or RAP with tram, where these NF κ B factors were not silenced, these data might indicate an alternative interpretation.

The authors state that knocking down Toll-1, Toll-9, dl, or dif did not significantly affect tumour-induced lethality in the absence of trametinib. However, this observation lacks a direct comparison of survival rates between animals with RNAi expression and those without RNAi treatment. Considering the data in Figure 1A in which animals with RAP tumours exhibit significantly better survival rates compared to those with silenced Toll or NF- κ B genes, this observation remains a potential contradiction to the conclusions.

Furthermore, the findings related to the administration of NF- κ B inhibitors indicate that they did not mitigate drug resistance in RAP tumours (Supplementary Figure 2). One of the

inhibitors even intensified host lethality in RAP tumours treated with tram. These results suggest that the overall model may benefit from re-evaluation or clarification.

The use of "Interestingly" in this context seems somewhat misplaced, as it raises the question of how a negative result can be deemed interesting, particularly since it does not clarify the potential of pharmacologically targeting NF- κ B in human patients with CRC.

Addressing these points should enhance the clarity and significance of the findings.

Regarding the assessment of survival, further clarification is necessary. The formula used for calculating survival is unclear. Would it not be simpler to use the proportion of transgenic flies relative to non-transgenic flies and multiply by 100? It appears that a formula involving multiplying both groups is being utilized. Is that the right approach?

Moreover, the researchers measured pupae but reported their findings in terms of adult survival, which could be misleading if there is late pupal lethality involved. Wouldn't it be more more precise to quantify adult flies?

Despite these issues, the revised manuscript reports key findings on the role of Wnt and the glucuronidation detoxification pathway in drug resistance in RAP tumours. This is nicely supported by utilising 'patient avatar' models and human CRC cells. The study offers strategies such as Wnt inhibition and identifies a set of genes that may serve as promising biomarkers for predicting patient responses. This research lays the groundwork for improved treatment approaches.

WNT Signalling Promotes NF- κ B Activation and Drug Resistance in KRAS-Mutant Colorectal Cancer- Response to EMBOJ reviewers for new manuscript EMBOR-2025-62365-T

Referee #1:

The authors addressed all my concerns in a satisfactory way, and the MS is now ready for publication.

Referee #2:

*Major comments 1 & 3: The results showing that NF- κ B signaling affects key glucuronidation and pentose pathway enzymes, leading to decreased trametinib glucuronidation levels, is clear. However, the extent to which NF- κ B signaling plays a role in trametinib efficacy and survival is still unclear in this model... There is still a major discrepancy in the data between the pharmacological and genetic targeting of NF- κ B, as these two methods lead to opposing results in trametinib-induced RAP lethality in vivo... In addition, the "inhibition of canonical NF- κ B activity by knockdown of *dl* or *dif* did not significantly suppress RAP tumour overgrowth in the presence of trametinib (compare Figure 142 1I-J to 1H, quantified in 1K)." ... The authors must acknowledge the discrepancies between the survival effects of genetic and pharmacological targeting of NF- κ B in the context of trametinib resistance. Proper discourse on this matter may be sufficient to include these confounding results. Outside of this, the authors addressed most of our questions.*

Response

We agree and our writing over-interpreted a simple discrepancy between genetic reduction and systemic drug administration. Our subtle dominant genetic modifier approach (removal of a single functional *dl* copy) is effectively tumor-specific (the rate-limiting tissue regarding RAS/Wg) and non-toxic (*dl* is genetically recessive); pharmacological inhibitors strongly reduce DI across the entire organism, which is much more likely to be toxic: for example, strong systemic inhibition of NF- κ B affects not only tumour cells but also has broader physiological impact such as the immune system. Previous studies have reported that targeting NF- κ B in cancer through pharmacological means has yielded limited success, largely due to toxicity arising from NF- κ B's essential role in maintaining cellular homeostasis and innate immunity (Rajan Radha Rasmi et. al., 2020; Véronique Baud et. al., 2009). We now better address this point in the manuscript (highlighted in blue text in the manuscript):

Results section, line 142 now states: "However, inhibiting this pathway via knockdown of *dl* or *dif* did not significantly suppress *RAP* tumour overgrowth in the presence of trametinib (compare Figure 1I-J to 1H, quantified in 1K), suggesting that rescue is non-autonomous to the tumour."

Results section line 235 now states: "JSH-23 trend towards improved survival did not rise to significance while QNZ reduced survival: in addition to reducing NF- κ B pathway activity QNZ

also inhibits production of TNF- α (Tobe *et al*, 2003), suggesting that off-targets may contribute to reduced survival.”

Discussion section line 360 now states: “Pharmacological inhibition—through canonical NF- κ B pathway activity inhibitor JSH-23 or signalling inhibitor QNZ (EVP4593)—did not significantly suppress *RAP* tumour-induced lethality below toxic levels. This suggests that broad targeting of NF- κ B may be problematic from a whole body standpoint, likely reflecting the large number of processes controlled by NF- κ B activity.”

Major Comment 4: ...JSH, the NF- κ B inhibitor that showed no effect on trametinib activity in vivo, can modestly enhance trametinib efficacy in one colon cancer cell line, but has a trametinib-independent function in another colon cancer cell line. Specifically, in T84 cells, JSH has no effect on its own, and the authors show a modest yet inconsistent enhancement cancer cell death when used in combination with trametinib. These results generally support the conclusion of the paper. In contrast, JSH employment in SW620 cells led to greater cancer cell death than trametinib alone, and the combination of trametinib and JSH was not greater than JSH treatment alone. Thus, JSH is functioning to reduce cancer cell survival independently of trametinib in SW620 cells. A dose-response curve of JSH would reveal whether the drug concentration is responsible for these variations, since differences are seen at baseline as a single agent, between the various cell lines. Inclusion of a third cell line would help decipher which outcome is more reproducible / relevant in CRC. Regardless, these results suggest context-dependent activity of JSH in CRC and in relation to trametinib, perhaps explaining its lack of efficacy in vivo.

Response:

In our *Drosophila* studies, we assessed drug resistance by monitoring fly survival. Systemic inhibition of NF- κ B did not lead to a significant increase in survival, likely due to compound toxicity, although we did observe a trend toward improved survival (Figure S3I). In contrast, this concern is absent in cell culture experiments. Consistently, in T84 cells, JSH-23 enhanced the anti-tumour effect of trametinib. Cancer cell lines harbour a large number of genetic alterations beyond KRAS and APC that influence drug sensitivity, which may explain the differential responses observed between the two cell lines tested. We agree that testing a panel of cell lines would be optimal, but feel this is beyond the scope of the manuscript.

Minor Comment 1: The authors added a sentence broadly describing the pathways that were investigated in their initial screen. They did not mention any specific targets, or show any additional data. Supplying this information should have been easy, as they already have the data available.

Response:

To initiate this study on what we felt was a fundamental and ongoing challenge in the CRC field, we focused on systematically screening candidate signalling pathways that might contribute to trametinib resistance in RAP tumours: this entailed a

limited/preliminary screen with limited repeats. For example, we initially examined the role of ABC transporters, which are frequently implicated in mediating drug efflux and chemoresistance. However, our data indicated that ABC transporters do not significantly influence drug response in the RAP tumour model, although we cannot discount redundancy (data not shown). We then expanded our screen to include additional pathways—such as JNK, NF- κ B, and autophagy—that have also been associated with tumour drug resistance in various contexts.

Referee #3

The critical issue is the genetic and pharmacological impact of dif-RNAi or dl-RNAi. While it is understandable that determining the effects of host lethality without influencing tumorigenesis may be challenging at this stage, the implications of dif/dl upregulation and the role of TNF α discussed in relation to pharmacological inhibition remain vague. Further exploration of these aspects could offer valuable insights into potential new avenues for investigating resistance to trametinib.

Specifically, the issue of survival remains. In Figs 1 and 2 (e.g., 1C, 2A compared to 1A), it is evident that both dif-RNAi and dl-RNAi in the RAP tumours resulted in decreased survival compared to RAP alone (or RAP with tram) -it would be beneficial to compare the statistical significance of these groups. The impact of silencing Toll1 or Toll9 on host survival is less severe compared to the effects of silencing dif or dl. However, the data still pose similar interpretational challenges.

This discrepancy complicates the interpretation of the results, particularly since the tram+ RNAi did not achieve a survival level that allows for unambiguous interpretation of the interactions. As the degree of rescue conferred by tram treatment with both dl-RNAi or dif-RNAi does not (seem) to differ significantly from that of RAP or RAP with tram, where these NF κ B factors were not silenced, these data might indicate an alternative interpretation.

The authors state that knocking down Toll-1, Toll-9, dl, or dif did not significantly affect tumour-induced lethality in the absence of trametinib. However, this observation lacks a direct comparison of survival rates between animals with RNAi expression and those without RNAi treatment.

Considering the data in Figure 1A in which animals with RAP tumours exhibit significantly better survival rates compared to those with silenced Toll or NF- κ B genes, this observation remains a potential contradiction to the conclusions.

Response:

To knockdown target genes, we often use different *Drosophila* strains, which can result in variable tumour induction efficiency. To minimize this variability, we applied temperature control as a regulatory measure. Importantly, all comparisons were made within the same experimental set—flies of the same genetic background, raised in parallel under identical temperature conditions. For example, Figure 1A were conducted at 27 °C, but Figure 1C and 2A were conducted at 29 °C. Elevated temperatures lead to more aggressive tumour development and a corresponding decrease in fly survival. Therefore,

the statistical comparisons are valid and not confounded by inter-strain or environmental differences.

Figures 1C and 2A should be compared to the control data shown in Figure S3F, rather than Figure 1A. Specifically, the survival rates at 29 °C are as follows: RAP+DMSO, 9.1%; RAP+trametinib, 8.9% (S3F); RAP+dif-RNAi+DMSO, 2.8%; RAP+dif-RNAi+tram, 35.9%; RAP+dl-RNAi+DMSO, 5.8%; RAP+dl-RNAi+tram, 42.7% (Figure 1C); RAP+Toll1-RNAi+DMSO, 21.39%; RAP+Toll1-RNAi+tram, 47.93%; RAP+Toll9-RNAi+DMSO, 12.24%; RAP+Toll9-RNAi+tram, 42.71% (Figure 2A). Across all conditions, the difference between the control and experimental groups remains below 13%, which falls within an acceptable range of variation and did not show significant differences.

Furthermore, the findings related to the administration of NF- κ B inhibitors indicate that they did not mitigate drug resistance in RAP tumours (Supplementary Figure 2). One of the inhibitors even intensified host lethality in RAP tumours treated with tram. These results suggest that the overall model may benefit from re-evaluation or clarification... The use of "Interestingly" in this context seems somewhat misplaced, as it raises the question of how a negative result can be deemed interesting, particularly since it does not clarify the potential of pharmacologically targeting NF- κ B in human patients with CRC. Addressing these points should enhance the clarity and significance of the findings.

Response:

Please see our response above. We revised manuscript as follows, including replacing the word "Interestingly" with clarifying text:

Results section, line 142 now states: "However, inhibiting this pathway via knockdown of *dl* or *dif* did not significantly suppress *RAP* tumour overgrowth in the presence of trametinib (compare Figure 1I-J to 1H, quantified in 1K), suggesting that rescue is non-autonomous to the tumour."

Results section line 235 now states: "JSH-23 trend towards improved survival did not rise to significance while QNZ reduced survival: in addition to reducing NF- κ B pathway activity QNZ also inhibits production of TNF- α (Tobe *et al*, 2003), suggesting that off-targets may contribute to reduced survival."

Discussion section line 360 now states: "Pharmacological inhibition—through canonical NF- κ B pathway activity inhibitor JSH-23 or signalling inhibitor QNZ (EVP4593)—did not significantly suppress *RAP* tumour-induced lethality below toxic levels. This suggests that broad targeting of NF- κ B may be problematic from a whole body standpoint, likely reflecting the large number of processes controlled by NF- κ B activity."

Regarding the assessment of survival, further clarification is necessary. The formula used for calculating survival is unclear. Would it not be simpler to use the proportion of transgenic flies

relative to non-transgenic flies and multiply by 100? It appears that a formula involving multiplying both groups is being utilized. Is that the right approach? Moreover, the researchers measured pupae but reported their findings in terms of adult survival, which could be misleading if there is late pupal lethality involved. Wouldn't it be more more precise to quantify adult flies?

Response:

In this study, we quantified the number of empty pupal cases, which indicates successful development into adult flies. So no need for concern—if pupal death occurs at later stages, it won't affect the interpretation, as relying on empty pupal cases avoids this kind of misleading outcome. Regarding our statistics, we agree—and appreciate the correction—and correct the formula as follows:

Line 413 now states: “Transgenic flies and non-transgenic flies’ pupae (TP and non-TP (endogenous control)) respectively were counted for each test tube, and percentage of adult survival to control was calculated using the formula $[(TP / \text{no-TP}) \times 100]$.”

Dear Prof. Cagan,

Thank you for transferring your revised manuscript to EMBO reports. I now went again through the manuscript, the referee reports from The EMBO Journal and your p-b-p-response and consider the remaining points of referees #2 and #3 as adequately addressed. Referee #1 already supported the publication of the previous version of the manuscript.

Before we can proceed with formal acceptance, I have these editorial requests I ask you to address in a final revised manuscript:

- Please provide the final manuscript text as a .docx formatted file (including legends for main figures, EV figures and tables - see below), but without the figures included. Figure legends should be compiled at the end of the manuscript text.
- Please provide individual production quality figure files as .eps, .tif, .jpg (one file per figure), of main figures and EV figures. Please upload these as separate, individual files upon re-submission.

The Expanded View format, which will be displayed in the main HTML of the paper in a collapsible format, has replaced the Supplementary information. You can submit up to 6 images as Expanded View. Please follow the nomenclature Figure EV1, Figure EV2 etc. The figure legend for these should be included in the main manuscript document file in a section called Expanded View Figure Legends after the main Figure Legends section. Additional Supplementary material should be supplied as a single pdf file labeled Appendix. The Appendix should have page numbers and needs to include a table of content on the first page (with page numbers) and legends for all content. Please follow the nomenclature Appendix Figure Sx, Appendix Table Sx etc. throughout the text, and also label the figures and tables according to this nomenclature.

In this case, I would suggest to keep the 8 main figures and to combine the supplementary figures to have up to 6 EV figures. There are also 3 Tables mentioned. Please upload these separately as EV tables. Please name these "Table EV1" etc.. If these are large (more than one page), they might rather be datasets. In that case, please name and upload these as "Dataset EV1" etc. and put their legends on the first TAB of the respective excel sheets.

Finally, please make sure that all figure panels are called out separately and sequentially in the manuscript text, and that also Tables and Datasets are called out sequentially using the correct nomenclature. Presently, there are e.g. no separate callouts for Fig. 6D,E,F, Fig. 6 is called out before Fig 5A-E and Table S1 is called out after Table S2 and 3. Please check.

- We updated our journal's competing interests policy in January 2022 and request authors to consider both actual and perceived competing interests. Please review the policy <https://www.embopress.org/competing-interests> and update your competing interests if necessary. Please name this section 'Disclosure and Competing Interests Statement' and put it after the Acknowledgements section.

- Please add up to 5 keywords to the manuscript and order the sections like this, using (only) these names:

Title page - Abstract - Keywords - Introduction - Results - Discussion - Methods - Data availability section - Acknowledgements (please include here also the funding information) - Disclosure and Competing Interests Statement - References - Figure legends - Expanded View Figure legends

- The data availability section (DAS) is restricted to large, deposited datasets produced during the study. Please indicate here if and where primary datasets produced in this study (e.g. RNA-seq, ChIP-seq, structural and array data) have been deposited. If no primary datasets have been deposited, please also state this in this section (e.g. 'No primary datasets have been generated and deposited').

- Please provide ONE complete author checklist including the author information.

- Please check again that the number "n" for how many independent experiments were performed, their nature (biological versus technical replicates), the bars and error bars (e.g. SEM, SD) and the test used to calculate p-values is indicated in the respective figure legends (main, EV and Appendix figures). Please also check that all the p-values are explained in the legend, and that these fit to those shown in the figure. Please provide statistical testing where applicable. Please avoid the phrase 'independent experiment', but clearly state if these were biological or technical replicates. Please also indicate (e.g. with n.s.) if

testing was performed, but the differences are not significant. In case $n=2$, please show the data as separate datapoints without error bars and statistics. See also:

<http://www.embopress.org/page/journal/14693178/authorguide#statisticalanalysis>

If $n < 5$, please show single datapoints for diagrams. Moreover:

1. Please note that the legends for figures 1 is not provided in the sequential manner (legend for sub-figure 1C-F is provided before legend of figure 1B). This needs to be rectified.
2. Please note that the legends for figures 6 is not provided in the sequential manner (legend for sub-figure 6D is provided before legend of figure 6C). This needs to be rectified.
3. Please indicate the statistical test used for data analysis in the legends of figures 1A, C, D, E, F, G, K; 2A, B, 3A, C, D, E, F, H; 4B, C, D, E; 5A-E; 6A, B, D-F, G-I; 7I, J
4. Please note that in figures 1A, C, E, G; 2A, 3C, D, F, H; 4B, C, E; 5A-E; 6A, B, D, G, H, I; 7I there is a mismatch between the annotated p values in the figure legend and the annotated p values in the figure file that should be corrected.
5. Please note that information related to n is missing in the legends of figures 1A, C, D, E, F, K; 2A, B; 3A, C, D, E, F, H; 4B, C, D, E; 5A-E; 6A, B, D-F, G-I; 8A-C
6. Please note that the error bars are not defined in the legends of figures 8A-C.
7. Please note that scale bar and its definition are missing for figures 2C-F.
8. Please note that the black arrows are not defined in the legend of figures 7D, H.

- Please add scale bars of similar style and thickness to microscopic images, using clearly visible black or white bars (depending on the background). Please place these in the lower right corner of the images themselves. Please do not write on or near the bars in the image but define the size in the respective figure legend. Presently, error bars are often rather small, are completely missing or have text nearby. Please check.

- Please make sure that all the funding information is also entered into the online submission system and that it is complete and similar to the one in the acknowledgement section of the manuscript text file.

- Please remove the instructions from the Reagents and Tools Table.

- Please provide all the requested source data. I attach again the source data checklist sent to you by our source data coordinator. Please upload the source data as one folder per main figure, grouping together all the files for one figure (and ZIPed together). Please also upload the completed checklist with your submission. If you want to provide further source for the EV figures, please upload these as one single folder with all source data in separate folders for each figure ZIPed together.

In addition, I would need from you uploaded separately:

I look forward to seeing the final revised version of your manuscript when it is ready. Please let me know if you have questions or comments regarding the revision.

Best,

All editorial and formatting issues were resolved by the authors.

Prof. Ross Cagan
University of Glasgow
School of Cancer Sciences
United Kingdom

Dear Prof. Cagan,

I am very pleased to accept your manuscript for publication in the next available issue of EMBO reports. Thank you for your contribution to our journal.

Kind regards,
